# Linking drought indices to impacts to support drought risk assessment in Liaoning province, China

Yaxu Wang[1,2,3], Juan Lv[1,2], Jamie Hannaford [3,4], Yicheng Wang[1,2], Hongquan Sun[1,2], Lucy J. Barker[3], Miaomiao Ma[1,2], Zhicheng Su[1,2], Michael Eastman[3]

[1]China Institute of Water Resources and Hydropower Research, Beijing 100038, China

[2]Research Center on Flood and Drought Disaster Reduction of the Ministry of Water Resources, Beijing 100038, China

[3]The UK Centre for Ecology & Hydrology, Oxfordshire, OX10 8BB, UK

[4]Irish Climate Analysis and Research UnitS (ICARUS), Maynooth University, Dublin, W23 F2K8, Ireland

*Correspondence to*:Juan Lv (lujuan@iwhr.com)

**Abstract.** Drought is a ubiquitous and recurring hazard that has wide ranging impacts on society, agriculture and the environment. Drought indices are vital for characterising the nature and severity of drought hazards, and there have been extensive efforts to identify the most suitable drought indices for drought monitoring and risk assessment. However, to date, little effort has been made to explore which index(s) best represents drought impacts for various sectors in China. This is a critical knowledge gap, as impacts provide important 'ground truth' information for indices used in monitoring activities. The aim of this study is to explore the link between drought indices and drought impacts, using Liaoning province (northeast China) as a case study due to its history of drought occurrence. To achieve this we use independent, but complementary, methods (correlation and random forest analysis) to identify which indices link best to drought impacts for prefectural-level cities in Liaoning province, using a comprehensive database of reported drought impacts whereby impacts are classified into a range of categories. The results show that Standardised Precipitation Evapotranspiration Index with a 6-month accumulation (SPEI6) had a strong correlation with all categories of drought impacts, while Standardised Precipitation Index with a 12-month accumulation (SPI12) had a weak correlation with drought impacts. Of the impact datasets, 'drought suffering area' and 'drought impact area' had a strong relationship with all drought indices in Liaoning province, while 'population and number of livestock with difficulty in accessing drinking water' had weak correlations with the indices. The results of this study can support drought planning efforts in the region and provide context for the indices used in drought monitoring applications, and so enabling improved preparedness for drought impacts. The study also demonstrates the potential benefits of routine collection of drought impact information on a local scale.

## 1 Introduction

Drought is one of the most pervasive natural hazards, and can cause numerous and severe societal impacts. Drought impacts are mainly non-structural, widespread over large areas, and are often have a delayed onset in relation to the start of the drought event; therefore, it is challenging to properly define, quantify and manage drought (Mishra and Singh, 2010). There are a

number of 'types' of drought (Wilhite and Glantz, 1985), such as meteorological, agricultural, hydrological, social and ecological drought. Meteorological drought is defined as a deficit of rainfall for a period in respect to the long term mean (Houérou, 1996). As these rainfall deficits propagate through the hydrological cycle, the other drought types occur as deficits occur in river flows, soil moisture and groundwater. Eventually impacts become manifest on the environment and society. China has experienced numerous droughts, which have caused great impact in many sectors since the 1950s, especially in Liaoning province in the dry northeast of the country (Zhang, 2004). Liaoning province experienced a severe drought from spring 2000 to autumn 2001 which captured a large amount of attention from stakeholders and caused serious impacts as a result of the consecutive years of drought (Chen et al., 2016).

The costly nature of droughts means it is essential to plan and prepare for droughts proactively. Drought risk assessment is an essential prerequisite of this proactive approach (Wilhite, 2000;Wilhite and Buchanan, 2005), and provides the methods to predict the potential drought risk to society and the environment. Some risk assessment efforts focus primarily on the meteorological indices of drought, e.g. assessing the risk of a given severity of meteorological drought using historical precipitation data (Potopová et al., 2015). However, to adequately assess drought risk it is also necessary to characterise the consequences of drought occurrence, i.e. the impacts of drought on society, the economy and the environment (United Nations International Strategy for Disaster Reduction (UNISDR), 2009).

There are a wealth of drought indices in the literature (Lloyd-Hughes, 2014), however they have predominantly used for drought monitoring and early warning (e.g. Bachmair et al. 2016b) rather than drought risk assessment applications. The range of drought indices reflects the different types of drought which can be monitored, e.g., meteorological, hydrological and agricultural (Erhardt and Czado, 2017). Many indices, such as the Standardised Precipitation Index (SPI), can be calculated over different time scales. This enables deficits to be assessed over different periods, and can help monitor different types of drought. For example, shorter time scales, such as the SPI for three or six months are used for agricultural drought monitoring while SPI accumulations for 12 or 24 months are often applied to monitor hydrological droughts (Hong et al., 2001;Seiler et al., 2002). In China, many indices are used for drought monitoring, such as Palmer Drought Severity Index (PDSI), SPEI, SPI, China-Z index, relative soil moisture and remote sensing indices (Hong et al., 2001;Wang and Chen, 2014;Wu et al., 2012;Yanping et al., 2018). Li et al. (2015) found that serious drought events occurred in 1999, 2000, 2001, 2007 and 2009 in China using SPEI. Zhao et al. (2015) compared drought monitoring results between self-calibrating PDSI and SPEI in China with emphasis on difference of timescales. Wu et al. (2013) developed an Integrated Surface Drought Index for agricultural drought monitoring in mid-eastern China. Drought monitoring efforts in China tend to focus on meteorological and agricultural drought monitoring. Based on this and previous drought studies, the SPI, SPEI, soil moisture and NDVI were selected in this research to characterise meteorological and agricultural drought. The relationship between drought indices and drought impacts, established by statistical methods (e.g. Bachmair et al. (2016a)), can be used for drought risk assessment and appraisal of

vulnerability. Vulnerability is by its nature difficult to define and measure, but in effect, drought impacts are 'symptoms' of
drought vulnerability and provide a proxy for vulnerability appraisal by demonstrating adverse consequences of a given
drought severity (Blauhut et al., 2015a).
There are many different types of drought impacts affecting many aspects of society and the environment, but drought impacts
are rarely systematically recorded (Bachmair et al., 2016b). Some countries and regions have established drought impact
recording systems to analyse historical drought impacts. A leading example of this is the US Drought Impacts Reporter
(Svoboda and Hayes, 2011) which was launched as a web-based system in July 2005. More recently, the European Drought
Impact report Inventory (EDII) has been established (Stahl et al., 2016). Such databases are an important step forward, but the
information in them is necessarily partial and biased, as a result of being effectively crowd-sourced text-based information
based on 'reported' impacts from a range of sources (the media, grey literature, etc.). In contrast to many other countries, China
has a relatively complete and systematically assembled, quantitative drought impact information collection system. Data are
collected and checked at the county level by the Drought Resistance Department via a formalised network of reporters, who
collect information on drought impacts on agriculture, industrial economy, and water supply in every village. These data then
are fed up to the national government and held by the State Flood Control and Drought Relief Headquarters (SFDH). This
consistent collection of impact reporting provides a rich resource for drought risk assessment. However, impacts by themselves
are not fully instructive and to help inform risk assessment there is a need to understand their relationship with quantitative
drought indices.
Understanding the relationship between drought indices and drought impacts, and drought vulnerability, is a vital step to
improve drought risk management (Hong and Wilhite, 2004). However, whilst there have been many studies developing,
applying and validating drought indices, relatively few studies have assessed the link between indices and observed impacts.
Bachmair et al. (2016b) noted that this literature tended to be dominated by studies focused on agricultural drought, generally
linking indices like the SPI/SPEI and crop yield. Examples appraising multi-sectoral impacts are much sparser – there are
several recent studies in Europe, utilising impact reports from the EDII. Stagge et al. (2014) and Bachmair (2016b) used
drought impacts from the EDII, and various time scales of SPI, SPEI and streamflow percentiles. They found that the
relationships between indices and impacts varied significantly by region, season, impact types, etc. whilst Blauhut et al. (2015a)
and Blauhut et al. (2015b) developed a quantitative relationship between drought impact occurrence and SPEI using logistic
regression in four European regions. However, all four studies assumed drought impacts were only measured by the drought
impact occurrence (i.e. whether there was, or was not, an impact in a given month), the number of impacts or a combination
of both. This means that all drought impacts had an equal weight without considering the duration, intensity or spatial extent
of the individual impacts. In contrast, Karavitis et al. (2014) analysed drought impacts transformed into monetary losses to
measure drought impacts in Greece; however, it is challenging to transform all drought impacts into monetary units – especially
the indirect impacts of droughts.
In China, previous studies have also focused on agricultural drought risk assessment. Hao et al. (2011) applied the information
diffusion theory to develop a drought risk analysis model which used affected crop area to measure the drought disaster. Zhao
et al. (2011) established the relationship between drought frequency and simulated crop yield data in Henan Plain, and Jia et
al. (2011) used the water stress coefficient and duration to establish a drought index. Li et al. (2009) analysed the links between
historical crop yield and meteorological drought and established a meteorological drought risk index by combining the drought
frequency, intensity, yield loss and extent of irrigation. The drought index was found to explain 60-75% of the major crop yield
reduction. In drought impacts studies, Xiao-jun et al. (2012) collected annual drought affected area, damaged area, and annual
losses in food yield in nation level from China Water Resources Bulletins to explore the water management strategies during
droughts. In Hao et al. (2011), drought impacts were only measured by the affected crop area at the 10-day time step at the
county level. In our research, eight types of drought impacts are collected to measure drought impacts in at the city unit (i.e.
prefectural) level in Liaoning province, including the drought affected area, damaged area and yield loss, but also drought
impacts on humans, livestock and the agricultural economy.
In summary, previous studies have focused on linking impacts to only one characteristic of drought (such as intensity or
duration of occurrence) with most focusing on meteorological drought and agricultural impacts with little application of the
results to drought vulnerability assessments, with the exception of Blauhut et al. (2015a), Blauhut et al. (2016) and Hagenlocher
et al. (2019), for example. Here we link drought indices to drought impacts in 14 cities in Liaoning province, northeast China,
showcasing the use of the Chinese drought impact data from the SFDH. Using the drought impact-index linkage, we evaluate
drought vulnerability in Liaoning province and assess what factors affect drought vulnerability. A drought vulnerability
evaluation method that can be extended to other areas is then developed. The objectives of this paper are:
1.  To identify when and where the most severe droughts occurred between 1990 and 2013 in Liaoning province;
2.  To identify which drought indices best link to drought impacts in Liaoning province;
3.  To determine which city or area has higher drought vulnerability in Liaoning province; and,
4.  To ascertain which vulnerability factor or set of vulnerability factors have a higher contribution to drought

vulnerability, as quantified in objective 3.

**2 Materials**
**2.1 Study area**
Located in the northeastern of China, Liaoning province, comprised of 14 prefectural cities, has a temperate continental
monsoon climate with an annual average precipitation of 686.4mm, which is unevenly distributed both temporally and spatially
(Cai et al., 2015). Figure 1 shows the annual average rainfall across Liaoning, the south-east receiving on average more than
1000mm a year, whilst the north-west receives less than 500mm per year.

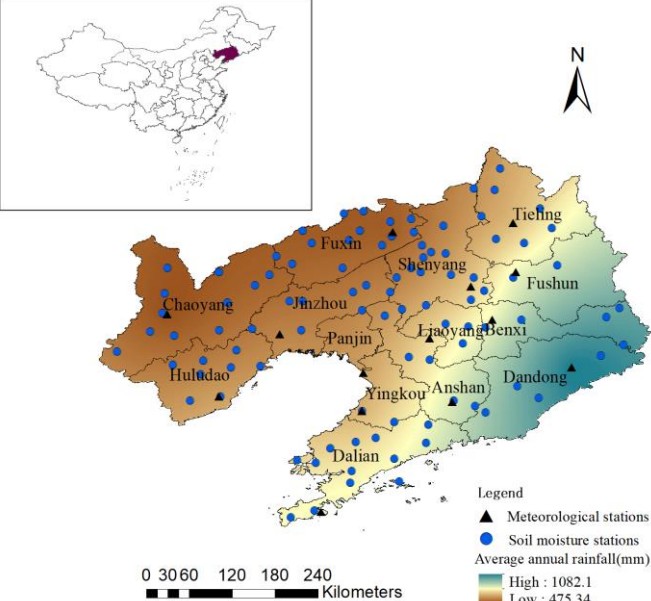


**Figure 1: Map showing the 14 prefectural cities, the distribution of meteorological and soil moisture stations and the average annual**
**precipitation in Liaoning province.**
The annual average volume of freshwater resources in Liaoning province is 34.179 billion $m^3$, and the annual average per
capita water resources is 769 $m^3$ – about one-third of the per capita water resources for the whole of China. Freshwater
resources are unevenly distributed within Liaoning province, with more freshwater resources in the south-east than the north-
west (Liu and Guo, 2009;Cao et al., 2012). Thus, Liaoning province is one of the provinces with severe water-shortages in
northern China. Liaoning province is also a highly productive area for agriculture; spring maize is the dominant crop in
agriculture production which makes it an important high-quality maize production area (Liu et al., 2013;Ren and Zhou, 2009).
Due to these characteristics, when drought occurs, as has frequently been the case in Liaoning province, it causes a significant
reduction in agricultural production (Yan et al., 2012). According to the SFDH, between 2000 and 2016 the average annual
yield loss due to drought was 1.89 million tons in Liaoning province, with an average annual direct agricultural economic loss
of 1.87 billion yuan.
**2.2 Data**
2.2.1 Meteorological data
Daily precipitation and temperature data for each city in Liaoning province for the period 1990-2013 were obtained from the
China Meteorological Administration (http://data.cma.cn/). Although there are 52 meteorological stations in Liaoning province,
due to the quality and length of the records, and location of the stations, one meteorological site in each city (shown in Figure
1) was selected to represent the meteorological condition for the whole city in order to derive drought indices.
2.2.2 Soil moisture data

Daily soil moisture data for 96 soil moisture stations in Liaoning province (shown in Figure 1) from 1990 to 2006 were obtained from Liaoning Provincial Department of Water Resources. Daily soil moisture was measured at three different depths: 10cm, 20cm and 30cm using frequency domain reflectometry soil moisture sensors Soil moisture data were not available at most stations between November and February due to freezing conditions.

2.2.3 Normalised Difference Vegetation Index (NDVI) data

Monthly MODIS Normalised Difference Vegetation Index (NDVI) data from 2000 to 2013 was collected in Liaoning province from the Geospatial Data Cloud (http://www.gscloud.cn/); the daily maximum data were used to derive the monthly average NDVI.

2.2.4 Impact data

In contrast to many other countries, China has a systematic, centralised drought impact information collection system. Drought statistics include drought impacts, drought mitigation actions and benefits of action to agriculture, hydrology and civil affairs. During a drought event, impact statistics are collected from every day to every three weeks, according to the drought warning level (Wang, 2014). When a drought warning is not triggered, drought impact data are collected after an event has ended which could be several months afterwards; and no data are collected when there is no drought event. Statistics for eight drought impact types were collected from the SFDH between 1990 and 2016, and aggregated to annual totals. The impact types used are listed in Table 1.

**Table 1: The eight drought impact categories for Liaoning province used in this study collected by the SFDH.**

| Impact | Abbreviation | Description | Unit |
|---|---|---|---|
| Drought suffering area | DSA | The area that was officially declared in drought | kha |
| Drought impacted area | DIA | The area that suffered crop yield loss by 10% or more | kha |
| Disaster area | DA | The area that suffered crop yield loss by 30% or more | kha |
| Recessed area | RA | The area that suffered crop yield loss by 80% or more | kha |
| Population with difficulty in accessing drinking water | PHD | Rural populations that cannot access normally to drinking water | 10k |
| Number of livestock with difficulty in accessing drinking water | NLH | Number of livestock that cannot access normally to drinking water | 10k |
| Yield loss due to drought | YLD | The amount of yield losses due to drought | 10k ton |
| Direct economic loss in agriculture | DELA | Direct losses of agricultural economy caused by drought | 0.1b yuan |

5) Vulnerability factors

The drought impacts described in Section 2.2.4 are mainly focused on agriculture sector. As a result of this, the availability of

data and the findings of Junling et al. (2015) and Kang et al. (2014), vulnerability factors relevant to these impacts were
selected. Vulnerability factors were collected from the 2017 Liaoning Province Statistical Yearbook to assess their contribution
to the drought vulnerability (Liaoning Province Bureau of Statistics, 2017), and are shown in Table 2 for each city unit.
**Table 2: Vulnerability factors for each city in Liaoning province collected from the 2017 Liaoning Statistical Yearbook (Liaoning**
**Province Bureau of Statistics, 2017)**

| City | Per capita gross domestic product (k yuan) | Population (10k) | Crop cultivated area (kha) | Annual per capita water supply (m³) | Per unit area of Fertiliser application (kg/ha) | Effective irrigation rate (%) | Number of electromechanical wells (k) | Reservoir total storage capacity (m m³) | Per unit area of major agricultural products (kg/ha) | Livestock production (10k ton) |
|---|---|---|---|---|---|---|---|---|---|---|
| Shenyang | 755.8 | 733.9 | 656.0 | 91.5 | 1000.4 | 40.0 | 27.6 | 686.6 | 7090.5 | 64.5 |
| Dalian | 1143.4 | 595.6 | 327.0 | 73.4 | 1437.2 | 22.8 | 19.0 | 2523.0 | 4914.3 | 70.8 |
| Anshan | 422.9 | 345.7 | 247.7 | 42.3 | 1031.8 | 30.1 | 4.1 | 91.9 | 6641.6 | 36.7 |
| Fushun | 402.7 | 214.8 | 116.1 | 94.7 | 776.9 | 37.4 | 1.8 | 2575.5 | 6342.9 | 10.4 |
| Benxi | 511.1 | 150.0 | 58.0 | 167.9 | 756.3 | 29.9 | 0.4 | 6078.8 | 6606.3 | 9.3 |
| Dandong | 315.8 | 237.9 | 190.4 | 28.0 | 1049.7 | 41.7 | 1.4 | 16202.8 | 6056.9 | 20.2 |
| Jinzhou | 341.8 | 302.2 | 457.2 | 46.6 | 915.4 | 41.3 | 18.7 | 977.9 | 6825.7 | 64.0 |
| Yingkou | 496.7 | 232.8 | 109.4 | 42.4 | 1564.6 | 67.7 | 12.3 | 269.6 | 7325.0 | 13.5 |
| Fuxin | 215.9 | 188.9 | 479.4 | 39.7 | 881.9 | 30.1 | 26.6 | 545.0 | 5243.6 | 49.6 |
| Liaoyang | 373.4 | 178.6 | 162.8 | 42.4 | 1002.6 | 44.8 | 4.0 | 1418.8 | 7202.2 | 11.0 |
| Panjin | 778.3 | 130.1 | 143.0 | 70.2 | 937.0 | 68.7 | 1.0 | 141.5 | 8918.3 | 23.8 |
| Tieling | 196.5 | 299.9 | 548.5 | 12.2 | 960.2 | 32.0 | 18.1 | 2174.5 | 8397.1 | 46.0 |
| Chaoyang | 210.1 | 341.1 | 464.5 | 15.8 | 874.7 | 42.0 | 17.4 | 2085.6 | 6292.0 | 63.6 |
| Huludao | 230.8 | 280.5 | 249.7 | 18.7 | 976.8 | 28.9 | 14.0 | 892.7 | 4852.3 | 35.4 |

**2.3 Methods**
2.3.1 Drought indices
Two meteorological indices were selected, the Standardised Precipitation Index (SPI; McKee et al., 1993) and the Standardised
Precipitation Evapotranspiration Index (SPEI;Vicente-Serrano et al., 2010). These standardised indices are widely used in
drought monitoring applications around the world, and the SPI is recommended by World Meteorological Organisation to
monitor meteorological drought (Hayes et al., 2011). This is due to the flexibility of being able to derive SPI over different
time scales and that it can be compared across time and space.
The SPI, in its default formulation, assumes that precipitation obeys the Gamma (Γ) skewed distribution, which is used to
transform the precipitation time series into a normal distribution. After normalisation, classes of drought can be defined with
the cumulative precipitation frequency distribution (Botterill and Hayes, 2012;Hayes et al., 1999). The SPEI uses the same
standardisation concept using the climatic water balance (that is, precipitation minus potential evapotranspiration; PET) instead
of precipitation. In this study PET is calculated by the Thornthwaite method (Thornthwaite, 1948), using observed temperature
and sunlight hours (estimated from latitude) as inputs. The SPEI is calculated by normalising the climatic water balance using
a log-logistic probability distribution (Vicente-Serrano et al., 2010).
SPI and SPEI are easily calculated and can fit a wide range of accumulation periods of interest (e.g. 1, 3, 12, 24, 72 months)
(Edwards, 1997). The SPEI has the added advantages of characterising the effects of temperature and evapotranspiration on
drought. In this study, SPI and SPEI were calculated for five accumulation periods, 6, 12, 15, 18 and 24-months, from 1990 to
2013 for 14 meteorological stations (i.e. one in each city – as shown in Figure 1). Generally, precipitation in Liaoning province
is concentrated between April and September which corresponds to the growing stage of spring maize. Considering the
climatology and crop growth period, SPI6 and SPEI6 ending in September were selected for this study, i.e. calculated using
precipitation during April to September. The 12, 15, 18 and 24 months SPI and SPEI in ending December were also analysed
with the annual drought impacts during 1990 to 2013.
Using the daily soil moisture of 10 cm, 20 cm and 30 cm depths, the daily average soil moisture for each station was calculated
using Eq. (1) and Eq. (2) (Lin et al., 2016).
$$\theta_1 = \theta_{10} \qquad \theta_2 = \frac{\theta_{10} + \theta_2}{2} \qquad \theta_3 = \frac{\theta_{20} + \theta_{30}}{2} \qquad (1)$$
$$\bar{\theta} = \frac{\sum_{i=1}^{3} (\theta_2 \times h_i)}{H} \qquad (2)$$
Where $\theta_i$ is the soil moisture of the $i$-th layer (i=1, 2, 3). $\theta_{10}$, $\theta_{20}$ and $\theta_{30}$ are the measured value at different depths
(10cm, 20cm and 30cm), $\bar{\theta}$ is the average soil moisture, $h_i$ is the thickness of the $i$-th layer of soil, and $H$ is the total
thickness of the measured soil.
Some of the daily soil moisture data were missing, however this was limited to 17% of the total soil moisture data. In some
cases there were missing data for one depth of soil moisture measurement. In these cases, the average soil moisture of the other
two layers was calculated, and where there was only one layer of soil moisture available it was used to represent the average
soil moisture. The annual average soil moisture was calculated based on the available daily soil moisture (March to October)
and was analysed with the annual drought impact data during 1990 to 2006. As each city has more than one soil moisture
station, the annual soil moisture of each station was calculated and then averaged to one value for each city.
The area-averaged NDVI at the city unit was calculated using the monthly NDVI. The critical stags of the spring maize growth
in Liaoning is in July, so the area-averaged NDVI in July was selected for the analysis with the annual drought impacts from
2000 to 2013.
2.3.2 Correlation analysis
The Pearson correlation method was used to characterise the correlation between indices and the selected drought impacts
(Özger et al., 2009). Due to the limited availability of soil moisture data, correlation analysis of soil moisture and drought
impact data was only carried out in 9 cities. The linkage between drought indices and impacts was used to assess the drought
vulnerability in Liaoning province. It can be inferred that the greater the impact caused by droughts at a specific severity
(measured according to SPI/SPEI), the higher the drought vulnerability of the city.
2.3.3 Random forest modeling
Decision trees are regularly implemented for machine learning tasks. They resemble flowcharts, consisting of a series of
branches, internal nodes, and leaf nodes. Internal nodes typically represent binary conditions of the explanatory variables.
These nodes are connected to other internal nodes by branches, which represent the outcome of the previous internal node.
Leaf nodes represent the outcome classes. Internal nodes are eventually connected to leaf nodes, which represent the outcome
classes of the classification task. Whilst quick to train and interpretable, decision trees are limited by overfitting to the training
set. Random forests (RFs) reduce overfitting by fitting an ensemble of uncorrelated decision trees. This is achieved using
bootstrap aggregation with replacement (bagging) and only considering a random subset of features for splitting at each internal
node (Breiman and Leo, 2001). As well as the reduction in overfitting compared to decision trees, the advantages of RF include:
its fast training speed, good accuracy and relative efficiency (Mutanga et al., 2012). Additionally, once RF models are
established, the values of the predictor that correspond to the first split in the decision tree can be extracted as thresholds
corresponding to impact occurrence (Bachmair et al., 2016a).
In this analysis random forests were built for regression. This is achieved by assigning categorical outcomes at each leaf node,
and using the mean prediction as the outcome. The R package 'randomForest' was employed to identify the relationship of
drought indices to drought impacts (Kursa, 2017;Liaw and Wiener, 2002) using 5000 decision trees for each RF model. The
variance explained was used to determine the goodness of fit of random forest model (Fukuda et al., 2013). The mean squared
error (MSE), Eq. (3), was used to evaluate the importance of each index:
$$MSE = \frac{1}{n} \sum_{i=1}^{i=n} (y_i - \hat{y}_i)^2 \tag{3}$$
Where $y_i$ and $\hat{y}_i$ are the observed drought impacts and the estimated drought impacts of each city, $i$, respectively; $n$ is the
length of time series.
The percent change of MSE (MSE%) is the difference in accuracy when the effect of the variable is excluded (i.e. if the SPEI6
is excluded from the model, the MSE% of the model may increase). Higher MSE% represents higher the index importance .
The first splitting values of each decision tree was also extracted.
Soil moisture and NDVI were not analysed using the random forest approach due to their short time series and prevalence of
missing data.
2.3.4 Standardisation of drought impacts and vulnerability factors
To ensure comparability and to facilitate the visualisation of the drought impacts and vulnerability factors they were
standardised to a value from 0 to 1 using Eq. (4) and Eq. (5) (Below et al., 2007).
$$SDI_i = \frac{DI_i - \min DI}{\max DI - \min DI} \qquad (4)$$
$$SVF_j = \frac{VF_j - \min VF}{\max VF - \min VF} \qquad (5)$$
Where $SDI_i$ and $DI_i$ are the Standardised Drought Impacts and the drought impacts of year $i$ in Liaoning province, respectively;
max $DI$ and min $DI$ are the maximum and minimum values of drought impacts in all year for the given impact type; $SVF_j$ and
$VF_j$ is the Standard Vulnerability Factors and vulnerability factors of city $j$ in Liaoning province; and max $VF$ and min $VF$ are
the maximum and minimum values of each category of vulnerability factors in all cities.
**3. Results**
**3.1 Drought monitoring and drought impacts**
Figure 2 shows the SPEI and the drought impact data for Liaoning province from 1990 to 2013; the *Sum of SDI* is the sum of
all types of Standardised Drought Impacts in the 14 cities for each year.

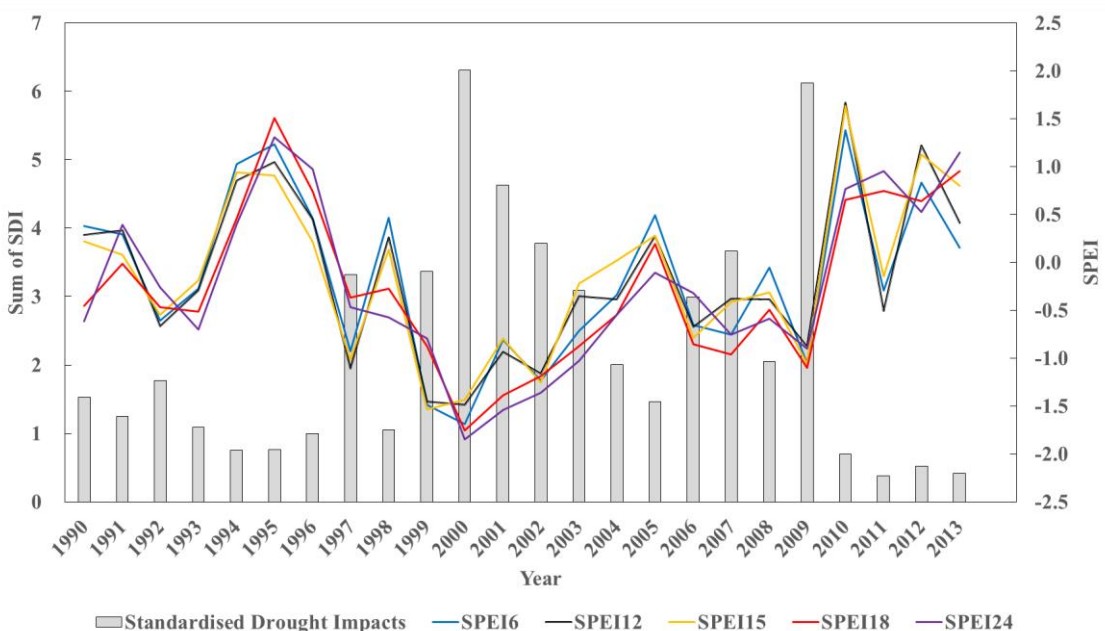


**Figure 2: Standardised Precipitation Evapotranspiration Index (SPEI) for 6-, 12-, 15-, 18- and 24-month accumulation periods and**
**the sum of the Standardised Drought Impacts (SDI) for each impact type listed in Table 1 for Liaoning province from 1990 to 2013.**
Figure 2 shows that the most severe droughts occurred in 2000, 2001, and 2009, whilst in 1994, 1995, 2012 and 2013 there
was above normal precipitation. From a visual inspection, the largest impacts are generally associated with the lowest index
values. This suggests that there is a relationship between the drought indices and drought impacts. This relationship is explored
quantitatively in the next sections.

Figure 3 shows the spatial distribution of the average of each drought impact type collected between 1990 and 2016. It shows that more severe drought impacts were recorded in the drier northwestern part of Liaoning province than in eastern parts of the province; the NLH was highest in Dalian, whilst Shenyang had the biggest PHD.

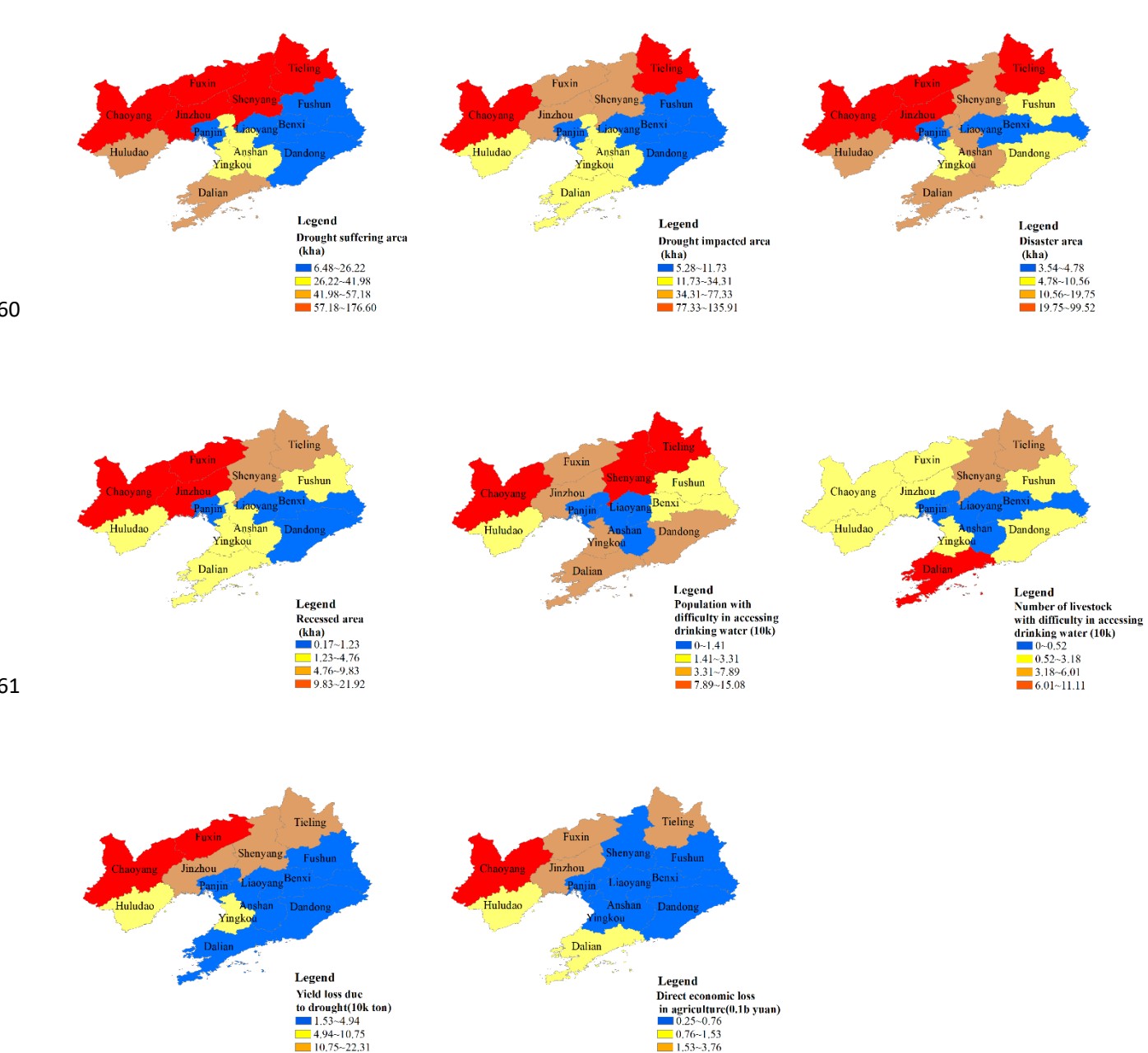

**Figure 3: Distribution of average drought impacts for each impact type, identified by the codes in Table 1) for the period 1990-2016 in Liaoning province.**

### 3.3 Correlation of indices with impacts

The Pearson correlation coefficient (*r*) for each city and drought impacts is shown in Figure 4. In most cases the drought index is negatively correlated with the drought impacts, suggesting that the lower (and more severe) the drought index, the greater drought impact. However, correlation strength, and direction, varied between the cities and impact types, ranging between -

0.890 to 0.621. In most cities of Liaoning province, NDVI and SoilM have a weak correlation with most of types of drought impacts. In Dalian, Chaoyang and Fuxin, all drought indices had a strong correlation with DA, whilst there was a significant correlation for drought impacts area in Jinzhou, Fuxin and Dalian, where most of the correlations were significant ($p < 0.01$). The strongest correlation was found between indices and PHD in Dalian, while it was the weakest in Dandong. There is a positive correlation between PHD and NDVI in Fushun, whist NLH has a positive correlation with NDVI in Anshan. Generally, SPEI6 had the strongest correlation with all types of drought impacts, whilst SPI12 had the weakest correlation. SPEI typically exhibited stronger correlations with drought impacts than SPI with the same accumulation period.

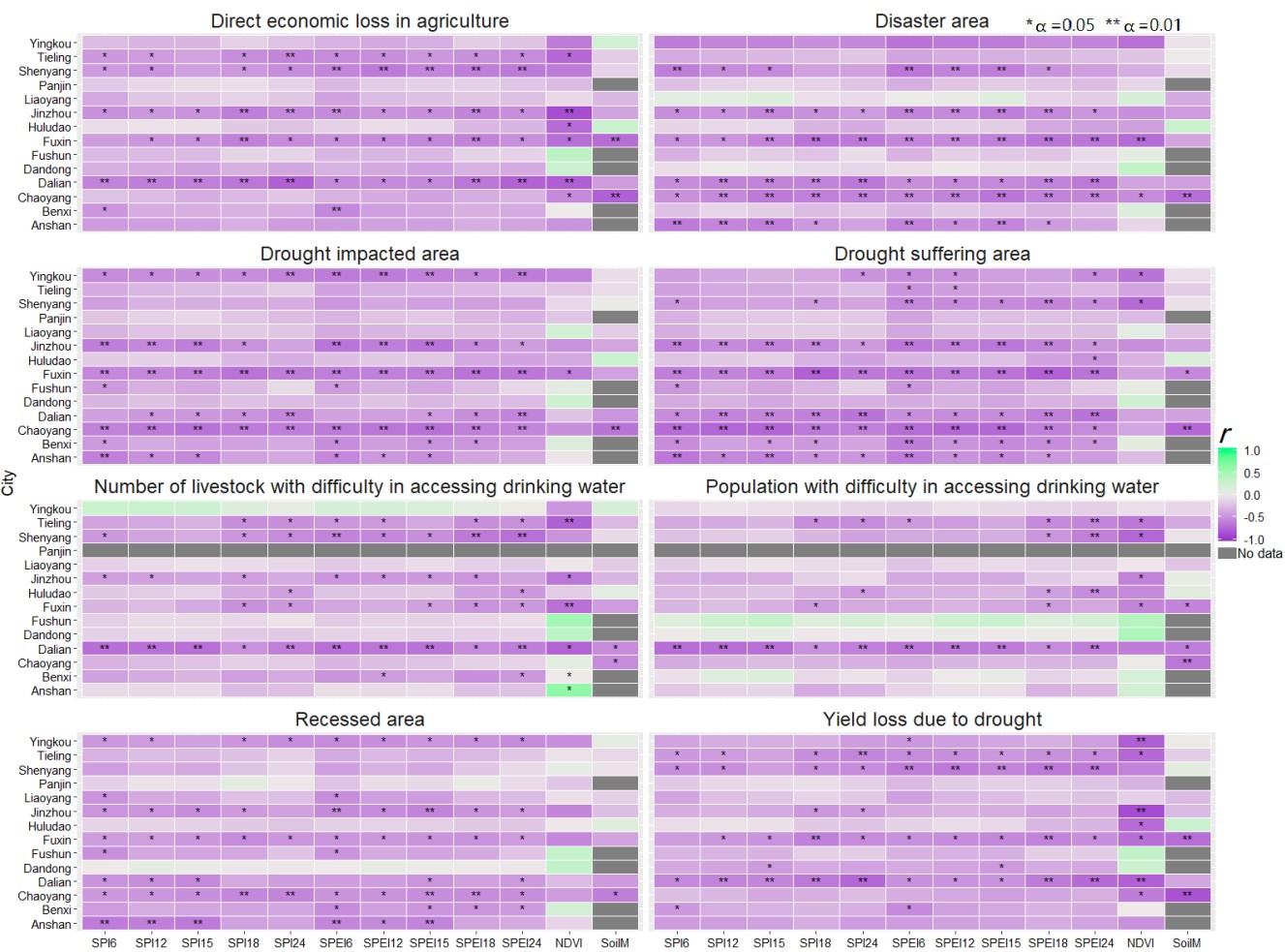

**Figure 4: Correlation coefficient (*r*) between drought indices (SPI, SPEI, NDVI and SoilM) and drought impacts for different impact types (identified by the codes in Table 1) in Liaoning province. The significance level of the correlation is indicated using asterisks.**

DSA and DIA had a strong correlation with all drought indices in Liaoning province, while PHD and NLH had a weak correlation. The average correlation coefficient across all drought indices and DSA in Liaoning was -0.43, while the average correlation coefficient with PHD and NLH was -0.22 and -0.27, respectively. Drought indices showed a moderate correlation with RA and YLD with average correlation coefficients of -0.32 and -0.37, respectively.

The performance of soil moisture varied significantly between cities and impact types (Figure 4); it had a strong correlation with the impacts in Chaoyang, and a weak correlation in Huludao. In Chaoyang, the correlation between soil moisture and

drought impacts was significant (α=0.01), whilst other cities were not significantly correlated.

## 3.4 Drought index importance in random forest models

Each drought impact type was selected as the response variable in the random forest. On average the random forests explained
41% of the variance observed within the drought impacts. The MSE% for each city and impact type is shown in Figure 5. The
MSE% can be seen to vary between different impact types. DIA and YLD have higher MSE% than other impact types, with
average MSE% is 3.02 and 3.01, respectively. The PHD and NLH had lower MSE%, with average of MSE% of 1.58 and 1.39,
respectively. DSA and RA had a moderate relationship with drought indices. SPEI performed better than SPI with same
durations; SPEI6 had the highest importance with drought impacts. SPI12 was the least important index in terms of drought
impacts. Indices had a higher importance with impacts in Anshan and Dalian and lower importance in Yingkou and Dandong.

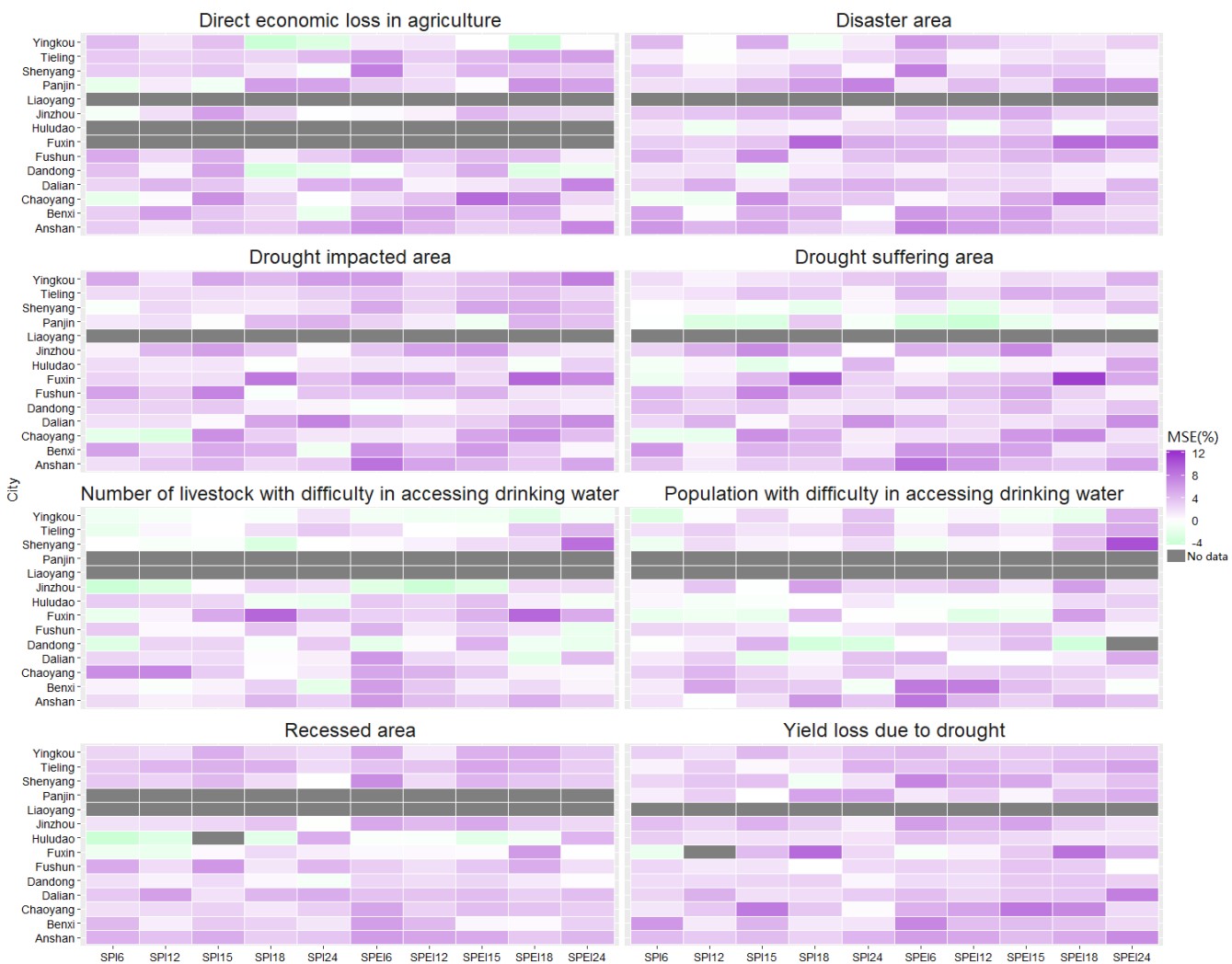

**Figure 5: The MSE% of drought indices (SPI and SPEI) with drought impacts (identified by the codes in Table 1) in Liaoning province using random forest.**

The variables identified MSE% from the random forest analysis generally match those with strong negative correlations. This
supports the statement that indices are negatively related to impacts. The threshold of impact occurrence based on the indices
was also identified in the RF analysis using the first splitting value. Figure 6 shows the distribution of first splitting values of

each decision tree within the RF. The average first splitting values for SPI18 and SPI24 were higher than those of SPI6, SPI12

and SPI15 (i.e. a more negative index value and more severe meteorological drought state) for all categories of drought impacts.

For SPEI, the results were similar (i.e.long-term deficits must be more severe to result in equivalent impacts compared to short-

term deficits) but there was more variability between accumulations. When viewed in terms of impact types, DSA had a low

threshold, indicating that DSA impacts occur more readily than DA or RA, as may be expected. The impact occurrence of

index values increase for DSA, DIA, DA and RA; and YLD and DELA tended to occur for more severe water deficits, with

the highest severity threshold being for NLH, indicating that only very severe drought conditions triggered impacts on livestock.

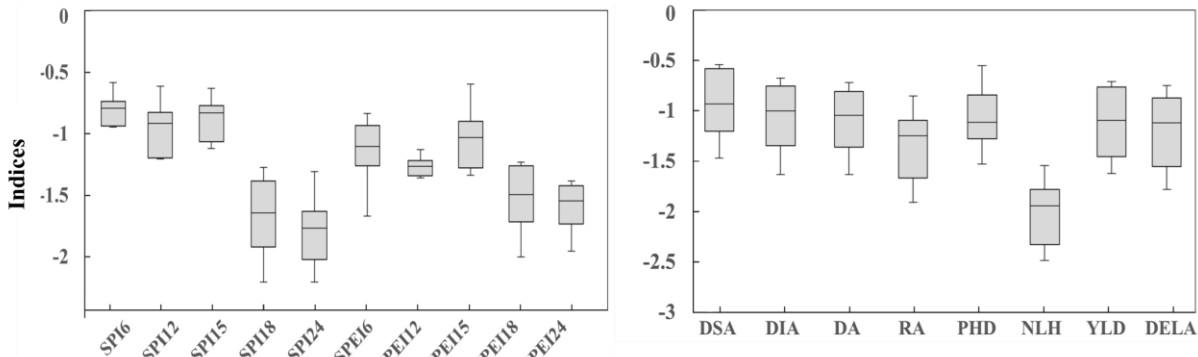

**Figure 6:**
**Box plots showing the splitting value (i.e. the thresholds of impacts) in the random forest construction across all impact types for each index (left), and across all indices for each impact type (right) in Liaoning province.**

**3.5 Drought vulnerability evaluation**

The results of correlation analysis and random forest suggest that in most parts of Liaoning province, SPEI at 6-month

accumulation period has the strongest relationship with drought impacts. SPEI6 was therefore selected to assess the drought

vulnerability of the 14 cities. Regression analysis was performed on the SPEI6 for each category of drought impact, and an

example is given in Figure 7 which shows the linear regression of DSA with SPEI6 in the 14 cities. It can be surmised that the

more serious the drought impacts for a specific drought severity (as defined by SPEI6), the higher the drought vulnerability.

Fuxin, Tieling, Chaoyang, Jinzhou and Shenyang have a higher vulnerability to DSA compared to the other cities.

Similar analyses were performed for all impact types, and Figure 8 displays the drought impacts each city in Liaoning province

is most vulnerable to. It can be seen from Figure 8 that there is little difference between cities in terms of sensitivity to various

categories of drought impacts. Considering the various impacts, Chaoyang, Jinzhou, Tieling, Fuxin and Shenyang had the

highest drought vulnerability - these cities are all located in the northwest part of Liaoning province. Dalian was most

vulnerable to NLH.

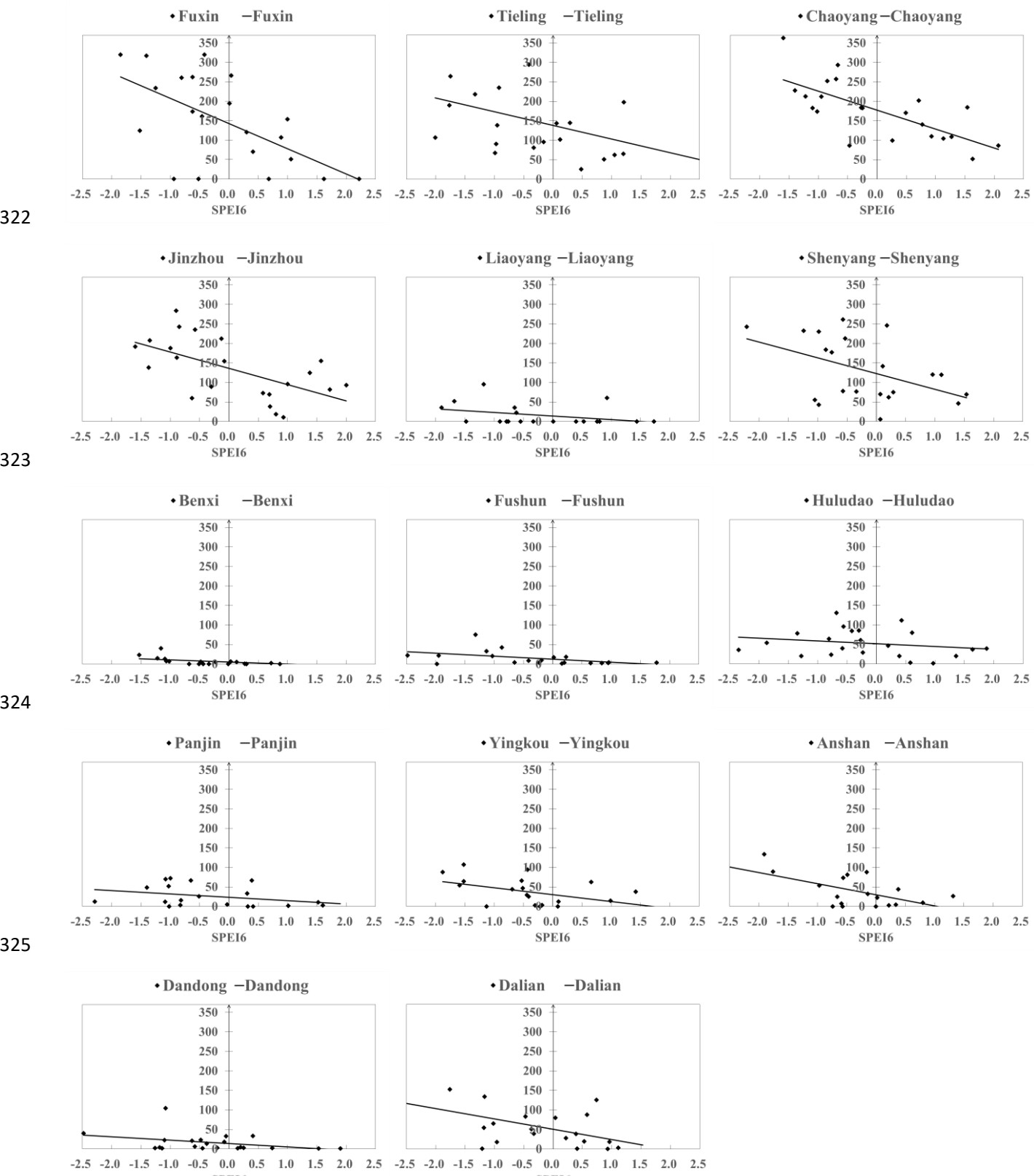






**Figure 7: Linear regression results of drought suffering area (DSA) with SPEI6 in each of the 14 cities in Liaoning Province.**

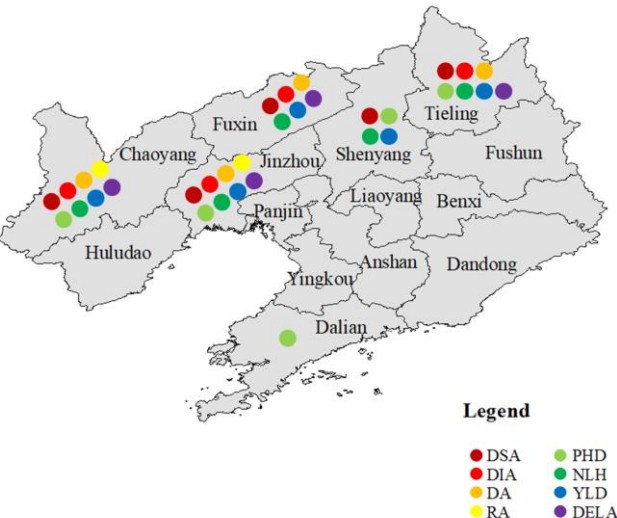


**Figure 8: Map showing the drought impacts each city in Liaoning province is most vulnerable to, based on the results of the linear regression.**

### 3.6 Vulnerability analysis

A stepwise regression model was built to explain the variation in each type of Standardised Drought Impact using vulnerability factors (listed in Table 2) as predictors. Drought impacts are symptoms of vulnerability and so can be used to estimate vulnerability to drought (Blauhut et al., 2015a). The vulnerability to drought can be assessed by maintaining a constant severity of drought (i.e. particular drought index values), and comparing the resultant impacts. More serious impacts correspond to higher vulnerability. Thus, the Standardised Drought Impacts corresponding to a severe meteorological drought (SPEI6 equal to -1.5), were regressed on the standardised drought vulnerability factors for 2017 to assess drought vulnerability for each drought impact type. Table 3 shows the results of stepwise regression model, demonstrating the contribution of vulnerability factors to each category of drought impact; the results varied for each impact type.

The relatively high $R^2$ values demonstrate the ability of the vulnerability factors to explain the variability exhibited by each drought impact. This is particularly the case for DSA, drought suffering area, and PHD, population with difficulty in accessing drinking water, which had associated $R^2$ values of 0.894 and 0.805, respectively. Population, crop cultivated area, and livestock production explained 89.4% of the variation in DSA. Population, in combination with per unit area of fertiliser application, and reservoir total storage capacity, also contributed to the DA model, explaining 80.5% of DA variation.

Population, crop cultivated area, and livestock production were identified as significant predictors in four, five, and three models, respectively, more than other vulnerability factors. Crop cultivated area, the most frequently significant predictor of drought impacts, also exhibited relatively high regression coefficients, demonstrating the strong relationship between the areas cultivated for crops, and the vulnerability to drought impacts. These results are paralleled in a composite drought vulnerability tool, which assigns relatively high weighting to area of land irrigated (Quinn, et al. 2014).

Population exhibited negative regression coefficients for three of four drought impacts, suggesting that as the population increased, the vulnerability to drought impacts decreased. However, population exhibits correlation with crop cultivated area, and livestock production. This, paired with potential unaccounted interactions between population and other predictors, may have resulted in inaccurate population coefficient estimation. This is supported by a positive population coefficient for predicting NLH. Population was used exclusively to predict NLH, thus correlation with other predictors, and interaction effects were unable to influence the coefficient. Furthermore, Figure 9 demonstrates that as population increases, DSA, DIA, and DELA increase. The composite drought vulnerability tool of Quinn et al. (2014) does not explicitly account for population, making a direct comparison not possible. However, it does assign a positive relationship between the ratio of rural population and drought vulnerability, which may explain the unexpected negative coefficients presented here. The number of electromechanical wells also exhibited a positive coefficient, suggesting that as the number of wells increases, drought impacts increase. However, it's possible that electromechanical wells are more prevalent in more drought prone areas, thus, the positive coefficient may simply demonstrate an association between electrometrical wells and RA.

Whilst drought vulnerability factors were able to explain 47.4-89.4% of the variability in drought impacts, annual per capita water supply, effective irrigation rate, and per unit area of major agricultural products were not identified as significant predictors of any drought impact type. It is important to consider, however, that correlation between these other vulnerability factors could result in them not being identified as significant, as the information is already contained within other vulnerability factors. This is supported by drought impact correlations with per capita water supply, and effective irrigation rate. However, minimal correlations between drought impacts and unit area of major agricultural products were observed, suggesting that the absence of a detected relationship may be a true reflection.

**Table 3: The vulnerability factors selected for the stepwise regression model and the $R^2$ of the resulting model for each impact type (identified by the codes in Table 1).**

| Drought Impacts | Predictors vulnerability factors) | Standardised Coefficients | $R^2$ |
|---|---|---|---|
| DSA | Crop cultivated area | 0.814 | 0.894 |
| | Population | -0.476 | |
| | Livestock production | 0.451 | |
| DIA | Crop cultivated area | 1.098 | 0.743 |
| | Population | -0.451 | |
| DA | Livestock production | 0.691 | 0.731 |
| | Per capita gross domestic product | -0.436 | |
| RA | Number of electromechanical wells | 0.629 | 0.541 |
| | Per capita gross domestic product | -0.452 | |
| PHD | Crop cultivated area | 0.949 | 0.805 |
| | Reservoir total storage | 0.472 | |
| | Per unit area of Fertiliser application | 0.352 | |
| NLH | Population | 0.720 | 0.474 |

| | | | |
|---|---|---|---|
| YLD | Crop cultivated area | 0.798 | 0.606 |
| DELA | Crop cultivated area | 0.556 | |
| | Population | -0.879 | 0.786 |
| | Livestock production | 0.793 | |

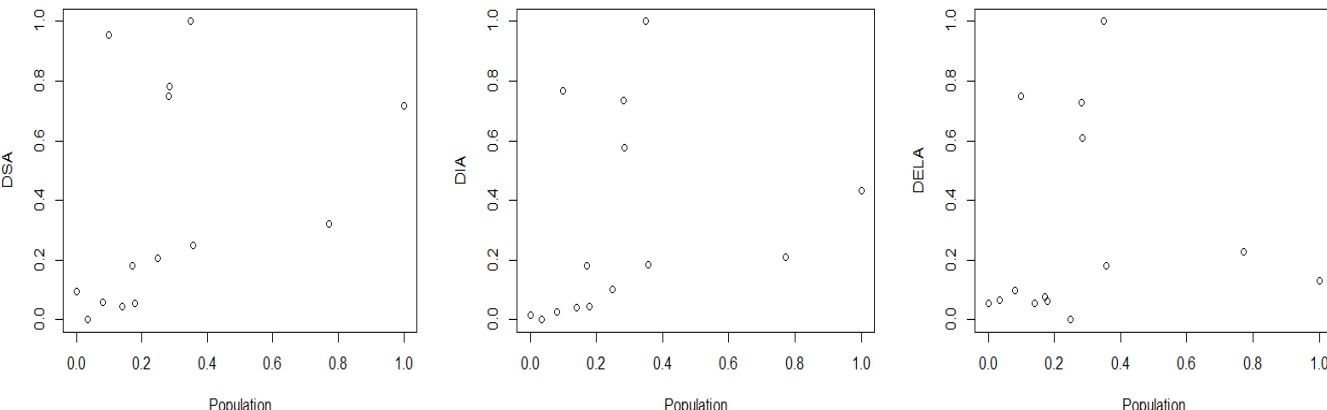

**Figure 9: Scatterplots demonstrating the association between population and drought impact in Liaoning province.**

## 4 Discussion

The methodology in this research has a number of distinctive characteristics in relation to previous drought impact and vulnerability assessments. The method takes many drought impacts, across a range of sectors, into consideration. The extensive drought impact data were systematically collected at county level, which is a consistent and reliable data source enabling regional comparisons. The drought impact data used here included impact variables that are rarely available in other settings: e.g. population with difficulty in accessing drinking water, number of livestock with difficulty in accessing drinking water, yield loss due to drought and direct economic losses in agriculture. In addition, we not only considered the occurrence of drought impacts, but also the severity of impacts and their spatial variation between regions. Finally, the relationship between drought indices and drought impacts was explored using different statistical approaches, and this linkage was used to assess drought vulnerability in Liaoning province using a range of vulnerability factors.

The study has some important limitations which must be considered in interpreting the outcomes. The biggest challenge was the spatial and temporal matching between the drought impacts and indices. The regularity with which impact data are collected is determined by the drought warning level and as such they are not evenly spaced in time; as a result of this, the data were aggregated to annual totals. It was important to match the accumulation period and timing of the selected drought indices to the timescales critical for the drought impacts; for example SPEI6 in September covers the critical maize growth period and is when the majority of precipitation falls. However, the results may change if we used multi-year drought impacts, as longer index accumulation periods may have a stronger correlation with multi-year drought impacts than single year drought impacts. Soil moisture data were collected at a daily resolution, in order to match up soil moisture and impact data, the March to October

average soil moisture was used in the correlation analysis. However, short term soil moisture deficits can have serious impacts on crops which are sometimes unrecoverable. The average soil moisture may not have captured these short-term deficits, particularly if soil moisture was, in general, sufficient the rest of the year. Also in some cities, the lack of soil moisture data means that the annual average soil moisture does not reflect the occurrence of typical agricultural drought during the year. For this reason, soil moisture data can be used for real-time drought monitoring applications, but may not appropriate to present drought impacts on an annual scale for risk assessment, as applied here. NDVI data for the critical growth period of spring maize was used in the analysis with annual drought impacts (i.e. July-month), but again this does not take all drought events during crop growing period into account. The correlation coefficients characterising the relationship between NDVI and drought impacts are both positive and negative; this is likely due to the complexity of NDVI drivers (e.g. diversity of land cover, crop types and growth stages etc.). For this reason, some studies have used the NDVI to identify the impact of drought on vegetation (Miao et al., 2018;Rajpoot and Kumar, 2018;Trigo et al., 2015;Wang et al., 2015).

The results from the correlation analysis were consistent with the results from the RF analysis. Drought suffering area (DSA) and drought impact area (DIA) had strong correlations with all drought indices in Liaoning province, while PHD and NLH have a weak correlation with indices. This was because DSA and DIA are direct impacts of agricultural drought, whilst PHD and NLH are related to many additional factors, such as drinking water source location and the quality of water resources; for example, livestock can drink water from the river directly, but the quality of the river water means it is not suitable for humans – for this reason, NLH showed least sensitivity to water deficits.

The random forest algorithms presented in this paper explained an average of 41% of the variance observed within the drought impact data. This is relatively modest, and may be partially due to limitation associated with the impacts data. The collinearity of the drought indices (e.g. SPI6 is correlated with SPEI6) is also a potential cause of the low MSE%. The correlation coefficients calculated for drought indices and NLH in Yingkou, and PHD in Fushun were positive. This was unexpected given the interpretation of these indices as estimations of the drought severity, and the majority of reported correlation coefficients being negative. Therefore, it seems likely this result is not representative of the true relationships between these indices and impacts, and instead is an artifact of imperfect impact data. To explore this, years with the highest numbers of impacts were removed before the correlation coefficients were estimated. This resulted in a negative correlation coefficient, providing further evidence for the positive correlation coefficients not being representative of the true relationships in these cities. The availability of more data would enable a better approximation of the true relationships between indices and impacts.

For all the drought impacts, Dalian and Fuxin showed the highest correlation coefficients among drought impacts and drought indices in all cases. The most vulnerable cities were Fuxin, Tieling, Chaoyang, Jinzhou and Shenyang, which are all located in the northwestern part of Liaoning province indicating there is a high drought vulnerability and drought risk in northwestern Liaoning. This is consistent with existing research by Yan et al. (2012) and Zhang et al. (2012), which established a drought

risk assessment index system to assess drought risk in northwestern Liaoning. Zhang et al. (2012) used indicators such as precipitation, water resources, crop area, irrigation capacity and drought resistance cost to measure drought risk and found that Fuxin, Chaoyang and Shenyang had a high drought risk.

The above results are also in general agreement with Hao et al. (2011), who used 10-day affected crop area data as the drought impacts to assess drought risk in China at the county unit. Their result showed that the West Liaohe Plain had a high risk. The results presented in our paper identify Chaoyang and Fuxin as having the highest drought vulnerability – the majority of these two cities are located on the West Liaohe Plain.

As the accumulation period increased, the first splitting value extracted from the random forest model tended to decrease, suggesting that at longer accumulation periods, larger water deficits are required for equivalent impacts to occur. There is a severe water deficits of RA occurrence since it caused more yield loss compared to DIA and DA. Drinking water for livestock requires lower water quality compared to that for humans, for example, livestock can drink water from the river directly, but the water quality of the river cannot meet the human drinking needs. For this reason, NLH showed least sensitivity to water deficits.

The relationships analysed in this research support the use of drought indices as a predictor of drought impacts and the impact thresholds identified can also support improved drought warning and planning. The drought vulnerability map (Figure 8) can be used to support drought risk planning, helping decision-makers to implement appropriate drought mitigation activities through an improved understanding of the drivers of drought vulnerability – for example, by sinking more wells to enhance resilience to drought (noting of course, that this measure has potential longer-term implications, for example, on groundwater exploitation e.g. Changming et al. (2001)).

The methods used here can be applied in other areas to better understand drought impacts and drought vulnerability, where similar data (e.g. drought impacts, meteorological data) are collected. While systematic, statistical archives of drought impact are comparatively rare, globally, there are numerous other potential sources of impact data that could be used (e.g. see Bachmair et al. 2016b).

**5 Conclusion**

This study used correlation analysis and random forest methods to explore the relationship between drought indices and drought impacts. It assessed drought risk in Liaoning province, and proposes a drought vulnerability assessment method which is applied to study the contribution of various socioeconomic factors to drought vulnerability. Here, we return to the original objectives of the study to summarise the key findings.

1. When and where the most severe droughts occurred between 1990 and 2013 in Liaoning province?

    Based on the drought monitoring results of SPI, severe droughts occurred in 2000, 2001, and 2009. In 2000-2001,

drought resulted in many impacts in Liaoning province, particularly in the northwestern part of Liaoning province.

The drought monitoring data showed corresponded well with the recorded drought impacts.

2.    Which drought indices best link to drought impacts in Liaoning province?

The results showed that the indices varied in their capacity to identify the different type of drought and impacts. The

strongest correlation was found for SPEI at 6 months, whilst SPI12 had a weak correlation with drought impacts.

SPEI was found to better link to drought impacts than SPI of the same accumulation period. NDVI and soil moisture

showed some links with impacts in some cities, but the results were generally weaker and less consistent than for

either SPI/SPEI – primarily reflecting the limitations in the soil moisture and NDVI datasets

3.    Which city or areas has a higher drought vulnerability in Liaoning province?

Chaoyang, Jinzhou, Fuxin, Shenyang and Tieling had higher drought vulnerability, all of which are located in the

northwestern part of Liaoning province, indicating that drought vulnerability is higher in these regions than in other

parts, which is consistent with previous research. However, in contrast with past work, the present research provides

a much more comprehensive assessment based on the occurrence of observed impact data.

4.    Which vulnerability factor or set of vulnerability factors have a higher contribution to drought vulnerability?

Population and crop cultivated area were strongly associated with drought vulnerability, suggesting these factors are

good indicators of drought vulnerability. However, the complexities of these relationships with drought vulnerability

require further investigation.

The results shown here give a clearer understanding about drought conditions in Liaoning province. The linkage developed
can be used to assess drought risk and to map vulnerability. It can also be used to help develop early warning systems and
predict drought impacts, which are vital tools for drought management. The results of the vulnerability analysis can guide
management measures to mitigate drought impacts – an important step to shift from post-disaster recovery to proactive pre-
disaster prevention.
**Data availability**
Some data, used during the study are proprietary or confidential in nature and may only be provided with restrictions (e.g.
drought impacts data, soil moisture). Daily meteorological data can be explored at http://data.cma.cn/ but access to data is
restricted. NDVI data can be obtained from the Geospatial Data Cloud (http://www.gscloud.cn/). Vulnerability data were
from the Liaoning Statistical Yearbook which can be obtained from Liaoning Province Bureau of Statistics
(http://www.ln.stats.gov.cn/).

**Author Contributions**

Yaxu Wang, Juan Lv, Jamie Hannaford, Yicheng Wang and Lucy Barker discussed and developed the aims of the paper. Yaxu Wang was responsible for the data analysis, visualisation and prepared the original manuscript, with contributions from Hongquan Sun, Lucy Barker, Jamie Hannaford, Miaomiao Ma, Zhicheng Su and Michael Eastman.

**Competing interests**

The authors declare they have no conflict of interest.

**Acknowledgements**

The authors gratefully acknowledge funding support for these researches provided by the National Key Research and Development Project (No. 2017YFC1502402), and Fund of China Institute of Water Resources and Hydropower Research (JZ0145B592016, JZ0145B582017) and China Scholarship Council. Jamie Hannaford, Lucy Barker and Michael Eastman were supported by the NERC National Capability Official Development Assistance project SUNRISE ("Sustainable Use of Natural Resources to Improve Human Health and Support Economic Development") [NE/R000131/1].

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
