# Peer review of "Linking drought indices to impacts to support drought risk"

_Natural Hazards and Earth System Sciences, 2019_

## Referee Comment (RC1) · Veit Blauhut (Referee) · 25 Oct 2019

Dear authors, congrates to your very well written piece of work exploring the unique Chinese drought impact database and establishing drought impact functions.Nice! In principal I agree with your study methodology and results. Nevertheless I do have some comments which you can find more detailed in the attached PDF.

The major points are you missed the publication of Hao et al. (2011) who used this kind of input in china- county level some caveats of defintiions/ clarification of terminology Please be more explicit explaining your vulnerability assessment: combination method etc.

[Figure]

Discussion: You almost fully miss a comparison to other studies (methodologies and results)

I'm looking forward to read over it again or discuss some of the actually existing, comparable approaches.
* * *
**Linking drought indices to impacts to support drought risk**
**assessment in Liaoning province, China**

Yaxu Wang[1,2,3], Juan Lv[1,2], Jamie Hannaford [3,4], Yicheng Wang[1,2], Hongquan Sun[1,2], Lucy J. Barker[3],
Miaomiao Ma[1,2], Zhicheng Su[1,2], Michael Eastman[3]

[1]China Institute of Water Resources and Hydropower Research, Beijing 100038, China

[2]Research Center on Flood and Drought Disaster Reduction of the Ministry of Water Resources, Beijing 100038, China

[3]Centre for Ecology & Hydrology, Oxfordshire, OX10 8BB, UK

[4]Irish Climate Analysis and Research UnitS (ICARUS), Maynooth University, Dublin, W23 F2K8, Ireland

*Correspondence to*:Juan Lv (lujuan@iwhr.com)

**Abstract.** Drought is a ubiquitous and reoccurring hazard that has wide ranging impacts on society, agriculture and the
environment. Drought indices are vital for characterizing the nature and severity of drought hazards, and there have been
extensive efforts to identify the most suitable drought indices for drought monitoring and risk assessments. However, to date,
little effort has been made to explore which index(s) best represents drought impacts for various sectors in China. This is a
critical knowledge gap, as impacts provide important 'ground truth' information. They can be used to demonstrate whether
drought indices (used for monitoring or risk assessment) are relevant for identifying impacts, thus highlighting if an area is
vulnerable to drought of a given severity. The aim of this study is to explore the link between drought indices and drought
impacts, using Liaoning province (northeast China) as a case study due to its history of drought occurrence. To achieve this
we use independent, but complementary, methods (correlation and random forest analysis). Using multiple drought indices –
Standardized Precipitation Index (SPI), Standardized Precipitation Evapotranspiration Index (SPEI), Soil Moisture (SoilM)
and the Normalized Difference Vegetation Index (NDVI) – and drought impact data (on crop yield, livestock, rural people and
the economy) correlation and random forest analysis were used to identify which indices link best to the recorded drought
impacts for cities in Liaoning. The results show that the relationship varies between different categories of drought impacts
and between cities. SPEI with a 6-month accumulation (SPEI6) had a strong correlation with all categories of drought impacts,
while SPI12 had a weak correlation with drought impacts. Of the impact datasets, 'drought suffering area' and 'drought impact
area' had a slightly strong relationship with all drought indices in Liaoning province, while 'population and number of livestock
with difficulty in accessing drinking water' had weak correlations with the indices. Based on the linkage, drought vulnerability
was analyzed using various vulnerability factors. Crop cultivated area was positively correlated to the drought vulnerability
for five out of the eight categories of drought impacts, while the total population had a strong negative relationship with drought
vulnerability for half the drought impact categories. This study can support drought planning efforts in the region, and
provides a methodology for application for other regions of China (and other countries) in the future, as well as providing
context for the indices used in drought monitoring applications, so enabling improved preparedness for drought impacts.

**Fig. 1.**

---

## Referee Comment (RC2) · Veit Blauhut (Referee) · 28 Oct 2019

[referee-annotated manuscript omitted]

---

## Referee Comment (RC3) · Anonymous Referee #2 · 29 Oct 2019

**General comments**

The paper is interesting, since it uses observed numerically measured drought impacts, which are often not available in many countries. The results clearly show that there is a correlation between drought indices and drought impacts. Vulnerability analysis is also appreciable, since it investigates the factors that make cities more vulnerable to droughts. The paper addresses questions within the scope of NHESS. I believe that some minor revisions are needed in order to make it publishable.

**Specific comments**

1. The abstract should be shortened; now it consists of 400 words, while NHESS standards foresee a 100-200 word abstract.

[Figure]

127: you said that one representative meteorological site in each city was selected to represent the meteorological condition for the whole city. Which are the criteria you adopted to select the representative station?

2. Line 135-136: NDVI data: you used MODIS data, which are available from 2000 to 2013. Why didn't you consider the NOOA AVHRR data, which span from 1981 to present? In this case you can include in your analysis also the period from 1990 to 2000.

3. Can you please specify how you computed the monthly average NDVI?

4. Line 140: which criteria is adopted to establish the beginning of a drought event or to trigger a drought warning according to the SFDH?

5. Line 244: Figure 2: since it seems that SPEI performs better than SPI, the same graph showed for SPI can be presented for SPEI too.

6. Line 234: Figure 3: it is not clear to me why at line 231 you say "Figure 3 shows the spatial distribution of the annual average of each drought impact type collected between 1990 and 2016" and in the figure caption you report a different period (1990-2013). Please, correct the wrong one.

7. Line 301-302: Can you please clarify why you select SPEI6=-1.5 for the second stepwise regression presented in the paragraph "Vulnerability analysis"?

**Technical corrections**

- Line 12: I believe there is a typing error: risk assessment (instead of risk assessments).

- Line 33-35: I would rephrase the sentence in the following way: "Drought is one of the most pervasive natural hazards which can cause huge societal impacts. Drought impacts are mainly non-structural, widespread over large areas, and delayed with respect to the event; therefore, it is still challenging to properly define, quantify and manage drought."

- Line 39: I will substitute "successive" with "consecutive".

- Line 60: I believe there is a typing error: impacts instead of impact.

- Line 66: I believe you forgot to insert from: "impacts from a range of sources..."

- Line 68: I believe there is a typing error: "at country level".

- Line 70-72: please review this sentence, since it is not clear.

- Line 80: I believe there is a typing error and "whilst" should not have a capital letter.

- Line 82-86: please review this sentence in order to explain better the concepts.

- Line 92: I believe there is a typing error: previous studies have BEEN focused.

- Line 115-116: please review the sentence "Thus, Liaoning province is one of the severe water-shortage provinces in northern China".

- Line 125 Remove "including daily precipitation and temperature"; you have already specified this point at the previous line.

- Line 147: Vulnerability factors were collected from the 2017 Liaoning province Statistical Yearbook to explain the drought vulnerability. Please, explain it better.

- Line 157: I believe there is a typing error: "The WMO recommends..."

- Line 162-166: Please, review the sentence, since it is not easy to understand.

- Line 170-171: I would change the sentence in the following way: "Precipitation in Liaoning province is concentrated between April and September; this period corresponds to the growing stage of spring maize".

- Line 171-172: Please review the sentence in order to explain which SPI6 and SPEI6 values you used in your analysis.

- Line 172-173: Please review the sentence in order to explain which SPI and SPEI 12, 15, 18 and 24 values you used in your analysis.

- Line 181: I will rephrase the sentence in this way "17

- Line 193-194: I will rephrase the sentence in this way "it can be inferred that the greater the impact caused by droughts of the same severity (measured according to SPI/SPEI), the higher the drought vulnerability of the city."

- Line 207: I believe there is a typing error "where $y_i$ and $\hat{y}_i$ are the observed drought impacts and the estimated drought impacts".

- Line 219: I believe there is a typing error: "... and min DI are the maximum..."

- Line 231-233: please, review the sentence to explain better what you have done.

- Line 277: I believe there is a typing error, since I cannot find an impact type called "DIS" in Table 1.

- Line 316: I would change the sentence in the following way: "data was systematically collected at country level".

- Line 239: I would change the sentence in the following way "but may not be appropriate..."

- Line 350: I would change the sentence in the following way: "Dalian and Fuxin showed the highest correlation coefficients among drought impacts and drought indices in all cases".

- Line 353-354: please rephrase the sentence.

- Line 356-360: please, rephrase the sentence, since it is not clear.

- Line 362: I would change the sentence in the following way: "The drought vulnerability map can be used to support drought risk planning, in order to help decision-makers to implement appropriate drought mitigation activities..."

- Line 372: I would substitute "severity" with "severe".

- Line 377: I would substitute "performance" with "perform".

- Line 387: I believe there is a typing error and "impact" should be used instead of "impacts".

---

## Referee Comment (RC4) · Doris Wendt (Referee) · 6 Nov 2019

The work of Wang, et al. presents not only an extensive and unique drought impact dataset, but it also shows how these impacts are linked to climate indices and how that relates to vulnerability to agricultural droughts in North East China. The presented case study shows that drought indices and impacts are linked throughout the region despite the large climate variability. Considering the novel application of linking these impacts and indices to a Chinese case study, I consider this work to be novel and its findings should be published in order to further ongoing drought research. However, I would suggest the following comments to improve the manuscript. My main suggestion is regarding the definition and use of drought terminology (see general suggestion 1 in attached PDF). Given the absent drought definition in the present manuscript, it is

somewhat difficult to compare this research with other drought studies in North East China. For example, the droughts identified in the first result section are not compared to meteorological or agricultural droughts of national scale. Relatively minor changes to the manuscript could emphasize the link between historical droughts and therefore increase the outreach of the research. The second section of the Results, the drought vulnerability results, shows that some cities are more vulnerable to agricultural drought than other cities. This result is extremely valuable and underlines the potential for furthering the developed methodology. Furthermore, the evaluation of drought vulnerability in Liaoning province also shows that the developed method can be applied despite large differences in climate. This also strengthens the general thought that the research is highly valuable and applicable to other regions in China. It is, however, not entirely clear how the vulnerability assessment would be applicable to other regions (given the available data). Improving this would strengthen the use of the developed methodology. Hence, a few minor changes in the phrasing of the term 'vulnerability' would aid to the general understanding of the study approach and applicability. Furthermore, adjustments regarding the vulnerability factors would strengthen current findings and align these with previous drought research.

Please find the attached PDF with general and specific comments.

Please also note the supplement to this comment:
https://www.nat-hazards-earth-syst-sci-discuss.net/nhess-2019-310/nhess-2019-310-RC4-supplement.pdf

**Supplement:**

**Review Wang, et al. 2019 –NHESS**

The work of Wang, et al. presents not only an extensive and unique drought impact dataset, but it also shows how these impacts are linked to climate indices and how that relates to vulnerability to agricultural droughts in North East China. The presented case study shows that drought indices and impacts are linked throughout the region despite the large climate variability.  Considering the novel application of linking these impacts and indices to a Chinese case study, I consider this work to be novel and its findings should be published in order to further ongoing drought research. However, I would suggest the following comments to improve the manuscript. My main suggestion is regarding the definition and use of drought terminology (see general suggestion 1 in attached PDF). Given the vague drought definition in the present manuscript, it is unnecessarily difficult to compare this research with other drought studies in North East China. For example, the droughts identified in the first result section are not compared to meteorological or agricultural droughts of national scale. Relatively minor changes to the manuscript could emphasize the link between historical droughts and therefore increase the outreach of the research.

The second section of the Results, the drought vulnerability results, shows that some cities are more vulnerable to agricultural drought than other cities. This result is extremely valuable and underlines the potential for furthering the developed methodology. Furthermore, the evaluation of drought vulnerability in Liaoning province also shows that the developed method can be applied despite large differences in climate. This also strengthens the general thought that the research is highly valuable and applicable to other regions in China. It is, however, not entirely clear how the vulnerability assessment would be applicable to other regions (given the available data). Improving this would strengthen the use of the developed methodology. Hence, a few minor changes in the phrasing of the term 'vulnerability' would aid to the general understanding of the study approach and applicability. Furthermore, adjustments regarding the vulnerability factors would strengthen current findings and align these with previous drought research.

**General suggestions**

The first suggestion is regarding the absence of a clear drought definition. The sentences in the Introduction [R36-37] are insufficient in describing what kind of 'numerous droughts' China has experienced, and how this study is related to drought studies in China or globally. The specific naming of the 2000-01 event and the frequent occurrence of drought [R118] calls for a rigid definition of a drought. Later in the Introduction, it only becomes clear that this study focuses on meteorological and (soil moisture) agricultural droughts. In my opinion, this should have been stated earlier and clearer. In addition to this, the manuscript gives little explanation of previous meteorological or agricultural drought events, even though multiple authors have described droughts in China on both national and regional level (Wu, et al. 2001; Zou, et al. 2005; Leng, et al. 2015, Xiao-jun, et al. 2012; Wang, et al. 2016). It would be beneficial to the manuscript to explore the link with previous studies and build on other national-scale drought studies to claim further implications of this study. For example, the presented dataset seems unique and unpublished, although the term 'China water resources bulletins' in Xiao-jun, et al. (2012) suggests that there are multiple sources of drought impact data. I would suggest that acknowledging of these relevant studies, as it helps to rightly place this new study in context of previous research and thereby support the claim of further implications of this study [R97-98 and R364-366]. In lines R364-366, it is stated that the method could be applied to other areas, although it remains unexplained how to do so. Results in Figure 3 and 4 suggests that the linking between drought impact data and climate

indices is fruitful despite the large climate variability. The results show the strong relation between SPEI6 and Drought suffering area (DSA), SPEI6 and drought impacted area (DIA), and yield reduction and NDVI. These relations could be explored further in the Discussion section (R364-366), if a rigid drought definition is applied and the findings are related to relevant studies. That would increase the outreach of the developed method and would therefore benefit the manuscript significantly. In other words, I would strongly recommend to 1) provide a definition of the studied drought events, 2) relate them to past events –strengthen objective 1- and 3) link the findings to other drought studies in China to show the relevance of this study. Given the current structure of the introduction, I would expect that these suggestions would strengthen both the first, second and sixth paragraph [R87-91].

In addition to suggestion 1, I would suggest to include relevant drought studies in China that have explored a meteorological index (Wu, et al. 2001), agricultural droughts (as referenced) and water resource management strategies (Xiao-jun, et al. 2012). The current overview given in paragraph 6 does not reflect the full spectrum of relevant studies, hence I would strongly suggest for a thorough review of relevant studies in China to emphasise the link between previous studies and these findings. These studies have also performed analysis using multiple sources of information and could therefore strengthen the second paragraph in the discussion R321-335.

The second suggestion concerns another definition; the use of the term vulnerability and the vulnerability assessment. In the Introduction, the relationship between drought indices, impact, and vulnerability is mentioned [R73-74], although in that same paragraph there is very little background given on the term 'drought vulnerability' or the chosen approach of this study. Later in the manuscript, R147-150, it becomes evident that vulnerability factors are related to agricultural productivity. It would strengthen the claim of 'developing a drought vulnerability evaluation' [R97], if the choice of vulnerability factors was justified earlier in the manuscript, perhaps supported using relevant literature to drought vulnerability.

The vulnerability factors themselves (Table 2) require some additional adjustment in my opinion. Currently, these factors do not relate to normal conditions, or below-normal conditions, i.e. drought conditions. The standardisation in R215-220 shows that vulnerability factors are a ratio that is relative to the maximum amount measured for an unknown time scale. It remains unknown how these factors are measured or would change over time and since these vulnerability factors are not given as a reduction from normal conditions, it remains unclear to the reader how they represent vulnerability. Without the full understanding of the vulnerability factors, the impact of Figure 8 is limited, as these vulnerability levels do not indicate vulnerability as such, solely a reduction from the maximum number. For example, it remains unclear what 'most vulnerable to' implies in Figure 8, and more explanation is required to understand which factors are in or excluded for which cities. If so, it would require some more explanation regarding the rationale behind these 'most vulnerable to' factors. Once the vulnerability factors are converted into a deviation from the long-term mean (or however a drought is defined), the combined effect of these factors would become clearer. I do not expect the results to change, although the factors will and potentially show the deviation from the mean (or normal) conditions and therefore emphasise the change during droughts. The results might show an amplified effects, which will help to strengthen the claim in R288-289. Along the same lines, I would also change the PHD, NLH and DELA into a percentage or ratio that relates to normal conditions. In the conclusion, relatively strong statements in R288-289 suggest that there is increasing drought vulnerability. However, from Figure 8 or Figure 7, it remains unclear how the vulnerability changes in Liaoning province, and these suggestions might aid the general analysis of the vulnerability factors.

The third suggestion is regarding the varying time scale of the multiple datasets. The presented data and analysis combine multiple datasets of varying quality and sources into one product. That in itself

is a fine bit of work, although I would suggest to show the applied time scale in the correlation analysis and in the random forest modelling. It is not a major concern, but it would strengthen the manuscript to frame a defined *study period* that matches all data analysed in the correlation analysis, i.e. 1990-2013. In R191-192 and in R211-212, a short statement is written regarding the limitations of the soil moisture data and the NDVI data. Perhaps, an additional note regarding the applied study period is best written here.

For consistency, I would also emphasise the applied time period for the random forest algorithm (as introduced in the third section of the Methods). In the current manuscript, the applied time period remains unknown for the Random Forest algorithm. In fact, to enhance clarity, a brief summary of the work of Bachmair, et al. (2016) would be beneficial for readers that are less familiar with this algorithm. Again, minor adjustments in the text would enhance the understanding of applied methods and therefore improve the manuscript.

Last correction I would suggest is the text along with Figure 5. In the figure, the coloured matrix gives the mean squared error in percentage. Firstly, I would strongly suggest to adjust the colour scheme to allow a non-experienced reader to see the difference between positive and negative percentage changes. Secondly, the change in MSE % suggests given a certain impact factor changes the error. If I read it correctly in R209, the change shows how much the accuracy decreases given the effect of the variable. This can be explained better than just one line of text, as a positive change in MSE% would imply *not* more MSE, but a *more accurate model*. Given the colour scale and the limited information available, the findings are somewhat hidden in this Figure despite the quality of the work. Hence, I would argue to change the colour scale accordingly and elaborate more in the text, i.e. give some examples.

**Specific comments**

The following section contains some specific comments to the text or figures:

There are four statements that specifically require an example to illustrate the statement:

- Regarding the aggregation of impact data to an annual time scale, I would suggest to dedicate a short paragraph in the Discussion [R340-349] to show if results change for a multi-year drought (2000-01) or for a one year drought (2009). You might be better placed to identify example drought events, but it would strengthen statements in R334-346.
- The NDVI results show both positive and negative correlations. In lines R334-335, it is stated that this could be due to diversity of land cover, but given the detailed vulnerability factors, I would assume that there could be a more elaborate answer to these correlations. It would strengthen the discussion section to highlight some of correlations to plausible explanation regarding, e.g. land cover, change of cropping, use of perennial crops, etc.
- Given the large spatial and temporal variability in precipitation [R108-110], it would be relevant to indicate the difference in water resources in addition to the variability in precipitation. The current annual average volume [R114-115] might not be relevant to drought conditions or vulnerability to droughts. The deviation from normal (annual average conditions) is relevant for drought research, how these droughts relate to the already water-stressed areas might be detected by the climate indices.
- The skewed distribution of water resources might play a part in the results of the DSA and DIA. It would be useful to indicate the deviation from mean, or the difference in source of water, rather than the amount that is available [R336-339]. In 358-360, the source and diversity of water sources is again linked to the vulnerability. This statement could benefit from an example case, where the source or variability in water resources indeed increased the vulnerability, as your results show.

The following specific comments are meant to increase the readability of the manuscript and might require a quick confirmation of the authors or adding of reference in order to avoid misunderstanding of terms.

- Change the layout of Table 1 so that the vulnerability factors are easier readable. This would shift the focus from being on the spatial variability (which would be better shown in a map than a table) to the different vulnerability factors.
- Depending on the applied drought definition (see general comment 1), mark this in Figure 2 to show the identified droughts. That will make it easier for the reader to deduct how the authors come to their findings in R128.
- Change the current volumes and amount in [0.1b] yuan of drought impact in percentages. For a reader that is not familiar with current production levels in Liaoning province, it is hard to grasp the loss of 1.89 million tons, or the impact of an economic loss 1.87 billion yuan when the normal conditions are not provided [R120-121].
- Repeat the abbreviations in Table 1 in the text and perhaps in Figure 3,4, and 5. The abbreviations are used throughout the result sections, but are only fully explained in Table 1. I would suggest to repeat the abbreviations in the text to enhance the readability. For example, include (DI) in R124 and (SDI) R218. Same for the vulnerability factors NLH and PHD [R223]. It would be better to first write them full, before abbreviating even though these are given in table 1.
- Need to support claims in drought mitigation strategies (e.g. sinking(?) more wells to enhance resilience to drought) R362-363.
- Could the authors clarify that the drought vulnerability map [R361] is indeed Figure 8?
- Other than in the abstract (R29-31), no findings are related to future applications for other regions in China. Please revise the abstract, as these statements can not be supported given the current manuscript.
- In R124 the meteorological data is introduced, I assume that this data is obtained from all stations in Figure 1, please indicate which stations were use, or refer to the figure in R124. The same holds for the soil moisture data in [R129].
- Explain the difference between the applied SPEI using the log-logistic probability distribution (Yu, et al. 2014) [R165-166] and the often used method of Vicente-Serrano, et al. 2010).
- Timeframe in R231 is 1990-2013 not 2016. Or, perhaps there is a mistake in the Figure 3 legend?
- Rephrase line 158-159
- Rephrase line 286-288
- Rephrase line 314-216
- Add 'of RF' in R356
- Rephrase line 358-360

**Additional references:**

Leng, G.; Tang, Q. & Rayburg, S. Climate change impacts on meteorological, agricultural and hydrological droughts in China  Global and Planetary Change, 2015 , 126 , 23 - 34

Wang, L.; Yuan, X.; Xie, Z.; Wu, P. & Li, Y. Increasing flash droughts over China during the recent global warming hiatus  Scientific reports, Nature Publishing Group, 2016 , 6 , 30571

Wu, H.; Hayes, M. J.; Weiss, A. & Hu, Q. An evaluation of the Standardized Precipitation Index, the China-Z Index and the statistical Z-Score International Journal of Climatology, 2001 , 21 , 745-758

Xiao-jun, W.; Jian-yun, Z.; Shahid, S.; ElMahdi, A.; Rui-min, H.; Zhen-xin, B. & Ali, M. Water resources management strategy for adaptation to droughts in China Mitigation and Adaptation Strategies for Global Change, 2012 , 17 , 923-937

Zou, X.; Zhai, P. & Zhang, Q. Variations in droughts over China: 1951--2003  Geophysical research letters, Wiley Online Library, 2005 , 32

---

## Author Comment (AC1) · 22 Dec 2019

Manuscript nhess-2019-310 "Linking drought indices to impacts to support drought risk assessment in Liaoning province, China" – Point by point response to referee 1 comments We thanks referee 1 for the feedback to our manuscript. We appreciate all the comments and suggestions and it is very useful to improve its quality and readability. We would like to address the referee's major concern. We have added a comparison with the paper of Hao et al. (2011), including methods, data and results. The vulnerability assessment method is a little confusing in the former manuscript, and we have added some explanations on how we did the quantitative vulnerability assessment. At the same time, some sentences are reorganized, which makes the manuscript more readable. In the discussion, we have added some comparison with other drought stud-

ies, including research data, methods and results. More detailed changes and replies are in bold below.

Line 43, That strongly differs to me 2016, Naumann 2018, Voigt 2019 and Hagenlocher et al 2019; Especially check for the latter–>drought risk and vulnerability review.

We thank the reviewer for this comment. Lots of drought risk assessment methods have been used in different study area. In Julia Urquijo Veit Blauhut 2016, some reviewed paper defined risk as follows.

$R = H \times V$

Where risk (R) is considered to be a function of hazard (H) and vulnerability (V).

It is similar to the one of the class in this manuscript which drought risk evaluate by drought hazard (drought frequency, severity etc.), vulnerability (including the drought resistance ability) and exposure of affected bodies (density of house, property and so on). Lots of indices were selected to measure the hazard, vulnerability and exposure. Drought risk is calculated by the weighted indices.

$R = H \times V \times E$

Where risk (R) is considered to be a function of hazard (H), vulnerability (V) and Exposure.

Another drought risk assessment method was grouped in this manuscript as follows.

Where risk (R) used to be considered to be a function of probability of drought (P) and potential consequences impacts (C).

In Blauhut et al. (2016), past drought impacts, drought indices and vulnerability factors were applied to assess drought risk. Blauhut et al. (2015) combine past drought impacts with hazard measurement in order to assess drought risk in pan-European. Probability of drought impact occurrence at five different drought hazard levels was used to measure drought risk. The higher of the probability of drought impact occurrence (potential consequences impacts) at same drought hazard levels (probability of drought), the higher of the drought risk. In Carrao et al. (2018) (Carrao, Naumann 2018), definitions of risk are commonly probabilistic in nature, referring to the potential impacts from a particular hazard in a future time period.

Essentially, these research assess drought risk from the drought severity and the potential drought impacts. These are similar with another class in this manuscript. We will clarify this in the revised manuscript..

Line 47: Indeed, quite some studies use 'risk' to characterize the hazard of drought (severity, frequency etc.). But the terminology of risk is by definition the likelihood of impacts! I recommend you to highlight this "missuse" a little more.

We agree with the reviewer and highlight this "missues" as suggested. Then we revised the manuscript.

Line 52: Reference?

Thanks for your suggestion and we added a reference here(Erhardt and Czado, 2017).

Line53: Vegetation drought?

We thank the referee for the comments and revised the manuscript to make it clear.

Line 66ïïjŽI believe Hao et al. 2011 published on this.

We thank the reviewer for this important comment. Hao's et al. (2011) is an important related study in this field to compare with. As mentioned above, we have added a reference to Hao et al in the revised manuscript. .

In Hao's study, drought impact only measured by affected crop area in a 10-day time step at county level, in our research, it is measured by eight types of annual drought impacts, which including the affected crop area at city level. Their result shows that West Liaohe Plain has a high risk. Most parts of Chaoyang and Fuxin are identified the highest vulnerability in our research which are located in West Liaohe Plain.

Line 78 ? Line 87 Line 94

We modified the reference as suggest.

Line 114: Does this mean from rainfall, recharge or water available for public water supply?

Here, it means the all the freshwater, we have clarified this in the revised paper.

Line 149: Please indicate why you selected these vulnerability factors: prestudies, expert knowledge, data availability, statistical tests?

We thank the reviewer for this comment and we fully agree with him on this point. Then, we added the reason why we selected these vulnerability factors. As mentioned in the response to reviewer 2, we selected these vulnerability factors, as the majority of impacts in Liaoning affected the agricultural sector. We have added some more description of this (and add some relevant previous studies that informed the selection) to the revised manuscript (Junling et al., 2015;Kang et al., 2014).

Line 170 SPI and SPEI are well know these days. Did you then interpolated between the stations, or did you keep stations values? If so why? Did you generalized to administrative border? If so how?

One station in each city was selected and calculated the SPI/SPEI for that one station and used it to represent the city (this meant the drought indices and drought impact data were at the same spatial scale – we will ensure this is clear in the revised paper.

Line 192 I just stumbled over this term again. You might explain what a city includes for your case. E.g. if you tale about cities in Europe it's only about the highly populated city centers, were actually no agriculture exists.

For this study, the city is divided by the city unit which include urban, town and village in its jurisdiction. We have added this definition to the data section.

Line 224 Where?

We changed the sentence and made clearer.

Line 227 How has this been done? Furthermore is this for a single city? Of are this averaged values? Please be more precise.

We thank the reviewer for this comment. We added a sentence to explain it clear on how we calculated the sum of SDI. "Sum of SDI is the sum of all types of Standardized Drought Impacts in 14 cities for each year." The standardization of the drought impacts is described in Section 2.2.4 of the methods.

Line 229 visually detected or proven by stats?

It is visually detected and we corrected and made clearer.

Figure 3 Please improve the resolution and use the full terms. How are the class defined.

We thank the reviewer for this comment and we fully agree with him on this point. We used the full terms and a higher resolution in the figure. The threshold is identified by the Natural Breaks (Jenks) method in the Arcmap.

Figure 4 Again, lots of space. present the full term please.

We agree with the reviewer and we present the full term as suggest.

Figure 6 Please increase resolution.

Yes we increased resolution as suggest.

Line 301 Please be more precise here vulnerability analysis is most often a big issue and quantitative approaches are lacking. Hence, readership might be highly interested here (at least I am).

We thank the reviewer for this important comment. We added some explanation on how do we quantitatively assess vulnerability.

Because for a specific severity of drought, basically, the more serious the impact

caused, the more vulnerable the region is. Thus, the regressed Standardized Drought Impacts at a moderate drought severity with SPEI6 equals -1.5 were applied to measure the drought vulnerability.

For example when SPEI6 is equal to -1.5, the regression result shows that yield loss due to drought is 5 thousand ton in Chaoyang whilst it is 1 thousand ton in Huludao. It means that in the term of the yield loss due to drought, Chaoyang is more vulnerable than Huludao.

Line 316 Please note that Blauhut et al. 2016 did similar, but on the basis of multivariable regression—checking for suitable hazard indices (SPI, SPEI, Soil moisture, VHI, CDI) and also checking trough a long list of vulnerability indices. Furthermore, the results of Hao et al. might be important for comparison.

The reviewer is right. We revised in the manuscript. Also we agree with the reviewer that we need make a comparison with Hao et al. 2011.

'The above results are also in general agreement with Hao et al. (2011), their study used 10-day affected crop area data as the drought impacts to assess drought risk in China at county unit. Their result shows that West Liaohe Plain has a high risk. Chaoyang and Fuxin are identified the highest vulnerability and most part of these two cities are located in West Liaohe Plain.'

Line 335 Here I strongly recommend a comparison to other studies which combined NDVI to impacts! Normally NDVI (or other vegetation health indices) suite very good.

We thank the reviewer for this important comment. We added some comparison with other studies which combined NDVI to impacts.

In other studies NDVI is mainly used to identify vegetation (agriculture) impacts. In this manuscript, affected human and livestock are also collected to measure drought impacts.

Line 339 How did other studies perform with respect to drought risk? In Liaoning

province? China? Globally? How good could teh linkage be detected by others? For agriculture?

We thank the referee for the suggestion that we included the comparison in the new version of the manuscript. We added some comparison with other studies which used different method of drought risk assessment.

Line 353: I suggest to prove this statically. Naumann et al. did an easy/brief stats on linking impacts and vulnerability in Africa. Or..you might consider to integrate vulnerability information in your risk model?

We thank the reviewer for this important comment. In Naumann et al. (2014), the tetrachoric correlation was computed between the drought vulnerability indicator and the numbers of persons reported affected to assess how the vulnerability indicators are correlated with drought disasters.

In our manuscript, stepwise regression model was built to compute between each type of Standardised Drought Impact and vulnerability factors to explore the contribution of the vulnerability factors to drought impacts. We will clarify it in the revised paper.

Line 354 This sentence feels a little loose here. What so you mean with this?

Yes, we delete this sentence.

Line 356 ?

Corrected.

Line 360 This is very detailed

We rephrased the sentence and revised the sentence to make it more readable.

Line 363 This is most often not a good idea! Besides that, the selected vulnerability factors limit management possibilities a lot. You might state that, the detection of drivers can support this

The reviewer is right. We changed the expression as suggested.

References

Blauhut, V., Stahl, K., Stagge, J. H., Tallaksen, L. M., De Stefano, L., and Vogt, J.: Estimating drought risk across Europe from reported drought impacts, hazard indicators and vulnerability factors, Hydrology and Earth System Sciences,20,7(2016-07-12), 20, 2779-2800, 2016.

Carrao, H., Naumann, G., and Barbosa, P.: Global projections of drought hazard in a warming climate: a prime for disaster risk management, Climate Dynamics, 50, 2137-2155, 2018.

Erhardt, T. M., and Czado, C.: Standardized drought indices: A novel uni- and multi-variate approach, Journal of the Royal Statistical Society, 2017.

Yan, L., Zhang, J., Wang, C., Yan, D., Liu, X., and Tong, Z.: Vulnerability evaluation and regionalization of drought disaster risk of maize in Northwestern Liaoning Province, Chinese Journal of Eco-Agriculture, 20, 788-794, 2012.

Zhang, J. Q., Yan, D. H., Wang, C. Y., Liu, X. P., and Tong, Z. J.: A Study on Risk Assessment and Risk Regionalization of Agricultural Drought Disaster in Northwestern Regions of Liaoning Province, Journal of Disaster Prevention  Mitigation Engineering, 2012.
* * *

---

## Author Comment (AC3) · 22 Dec 2019

Manuscript nhess-2019-310 "Linking drought indices to impacts to support drought risk assessment in Liaoning province, China" – Point by point response to referee 2 comments.

We thanks referee #2 for the feedback to our manuscript. We appreciate all the comments and suggestions and it is very useful to improve its quality and readability. The detailed Answer to each comment and suggestion are as follows.

Specific comments

The abstract should be shortened; now it consists of 400 words, while NHESS standards foresee a 100-200 word abstract.127: you said that one representative meteorological site in each city was selected to represent the meteorological condition for the whole city. Which are the criteria you adopted to select the representative station?

We thank the reviewer for this important comment. We made it shorter as suggested.

We considered the data quality, length of the time series and location of the stations to select the representative station. We have added some content to make it clearer.

Line 135-136: NDVI data: you used MODIS data, which are available from 2000 to 2013. Why didn't you consider the NOAA AVHRR data, which span from 1981 to present? In this case you can include in your analysis also the period from 1990 to 2000.

We thank the reviewer for this important comments. Yes, it is true that NOAA AVHRR has a long time series data. But, on the one hand, considered the advanced characteristics of MODIS data with high spatial and spectrum resolution, greatly improved data acquire ability and widespread applied on drought monitoring, etc., we used MODIS data instead. Also, it is easy to get access to the MODIS product of the NDVI.

Can you please specify how you computed the monthly average NDVI?

The monthly average NDVI products are available from the Geospatial Data Cloud (http://www.gscloud.cn/), and all the products quality were inspected. So we directly downloaded and used the monthly average NDVI products. According to the description of the products, the daily maximum NDVI data were used to represent the monthly average NDVI.

Line 140: which criteria is adopted to establish the beginning of a drought event or to trigger a drought warning according to the SFDH?

There are several criteria to trigger a drought warning for SFDH according to the local specific Drought Preparedness Plan, , such as meteorological drought monitoring result from the China Meteorological Administration, Soil moisture and hydrological drought monitoring from the Ministry of water resources and drought impacts collected

from the county-level in the system. So, basically, the criteria to trigger a drought warning are the integration of multiple factors, and are different with varied places.

Line 244: Figure 2: since it seems that SPEI performs better than SPI, the same graph showed for SPI can be presented for SPEI too.

We thank the reviewer for this comment and we fully agree with him on this point. We presented the SPI for SPEI in figure 2 as suggest.

Line 234: Figure 3: it is not clear to me why at line 231 you say "Figure 3 shows the spatial distribution of the annual average of each drought impact type collected between 1990 and 2016" and in the figure caption you report a different period (1990-2013). Please, correct the wrong one.

Thanks for your suggestion. We corrected this error.

Line 301-302: Can you please clarify why you select SPEI6=-1.5 for the second stepwise regression presented in the paragraph "Vulnerability analysis"?

Firstly, there are a certain amount of drought impacts when SPEI6=-1.5 for all types of drought impacts. If we use the result of regression analysis when SPEI6 is equal to -1, some types of the drought impacts are not be triggered. If we use the result of regression analysis when SPEI6 is equal to -2, due to the serious of drought, there could be serious drought impacts in all cities. Secondly, the results of relative value between cities are consistent when SPEI ranges from -1 to -2. Therefor we select SPEI6=-1.5 as a suitable point to stepwise regression.

Technical corrections

Line 12: I believe there is a typing error: risk assessment (instead of risk assessments).

Yes we corrected it and delete 's'.

Line 33-35: I would rephrase the sentence in the following way: "Drought is one of the most pervasive natural hazards which can cause huge societal impacts. Drought impacts are mainly non-structural, widespread over large areas, and delayed with respect to the event; therefore, it is still challenging to properly define, quantify and manage drought."

Yes we rephrased the sentence as suggest.

Line 39: I will substitute "successive" with "consecutive".

Yes we replaced the "successive" with "consecutive".

Line 60: I believe there is a typing error: impacts instead of impact.

Corrected.

Line 66: I believe you forgot to insert from: "impacts from a range of sources.."

Corrected.

Line 68: I believe there is a typing error: "at country level".

The data collection system reports to the national level through different grades of government, county-city-province-nation. Here it describes the data reporting process although this study obtains data from national level.

Line 70-72: please review this sentence, since it is not clear.

Rephrased.

Line 80: I believe there is a typing error and "whilst" should not have a capital letter.

We corrected it as suggest.

Line 82-86: please review this sentence in order to explain better the concepts.

We rephrased the sentence.

Line 92: I believe there is a typing error: previous studies have BEEN focused.

We corrected it and added 'been'.

[Figure]

Line 115-116: please review the sentence "Thus, Liaoning province is one of the severe water-shortage provinces in northern China".

We changed it as suggest.

Line 125 Remove "including daily precipitation and temperature"; you have already specified this point at the previous line.

We delete it as suggest.

Line 147: Vulnerability factors were collected from the 2017 Liaoning province Statistical Yearbook to explain the drought vulnerability. Please, explain it better.

We thank the reviewer for this comment we made it clearer.

Line 157: I believe there is a typing error: "The WMO recommends: : :"

Corrected.

Line 162-166: Please, review the sentence, since it is not easy to understand.

We have restructured the sentence to make it clear.

Line 170-171: I would change the sentence in the following way: "Precipitation in Liaoning province is concentrated between April and September; this period corresponds to the growing stage of spring maize".

Thanks for your suggestions. We replaced the sentence.

Line 171-172: Please review the sentence in order to explain which SPI6 and SPEI6 values you used in your analysis.

We agree with the reviewer and we added some explanation to make it clearer.

Line 172-173: Please review the sentence in order to explain which SPI and SPEI 12, 15, 18 and 24 values you used in your analysis.

Rephrased.

Line 193-194: I will rephrase the sentence in this way "it can be inferred that the greater the impact caused by droughts of the same severity (measured according to SPI/SPEI), the higher the drought vulnerability of the city."

Thanks for your suggestion and we replaced it as suggest.

Line 207: I believe there is a typing error "where yi and ˆ yi are the observed drought impacts and the estimated drought impacts".

Corrected.

Line 219: I believe there is a typing error: " and min DI are the maximum"

Yes we corrected this error.

Line 231-233: please, review the sentence to explain better what you have done.

Yes we have rephrased it.

Line 277: I believe there is a typing error, since I cannot find an impact type called"DIS" in Table 1.

We corrected this error.

Line 316: I would change the sentence in the following way: "data was systematically collected at country level".

We have replaced this sentence in the revised paper as suggest.

Line 239: I would change the sentence in the following way "but may not be appropriate.."

Rephrased.

Line 350: I would change the sentence in the following way: "Dalian and Fuxin showed the highest correlation coefficients among drought impacts and drought indices in all cases".

Thanks for your suggestions and we have replaced this sentence.

Line 353-354: please rephrase the sentence.

Thanks for your suggestions. We restructured the sentence to make it clear.

Line 356-360: please, rephrase the sentence, since it is not clear.

We rephrased the sentence to make it clearer.

Line 362: I would change the sentence in the following way: "The drought vulnerability map can be used to support drought risk planning, in order to help decision-makers to implement appropriate drought mitigation activities"

Thanks for your suggestions. We replaced with this sentence.

Line 372: I would substitute "severity" with "severe".

We replaced"severity" with "severe".

Line 377: I would substitute "performance" with "perform".

We have replaced "performance" with "perform" in the revised paper.

Line 387: I believe there is a typing error and "impact" should be used instead of"impacts".

Corrected.

References

Liaoning Province Bureau of Statistical: Liaoning Statistical Yearbook 2016, China Statistics Press, 2017.

Yan, L., Zhang, J., Wang, C., Yan, D., Liu, X., and Tong, Z.: Vulnerability evaluation and regionalization of drought disaster risk of maize in Northwestern Liaoning Province, Chinese Journal of Eco-Agriculture, 20, 788-794, 2012.

Zhang, J. Q., Yan, D. H., Wang, C. Y., Liu, X. P., and Tong, Z. J.: A Study on Risk Assessment and Risk Regionalization of Agricultural Drought Disaster in Northwestern Regions of Liaoning Province, Journal of Disaster Prevention & Mitigation Engineering, 2012.
* * *

---

## Author Comment (AC4) · 22 Dec 2019

Manuscript nhess-2019-310 "Linking drought indices to impacts to support drought risk assessment in Liaoning province, China" – Point by point response to referee 3 comments

We thanks referee #3 for the feedback to our manuscript. The comments and suggestions are particularly useful for us to revise the manuscript. Based on the suggestions, we have added the definition of drought and compared with other related studies. We emphasized the link between historical droughts studies and this manuscript, and added more explanations on how we did the quantitative vulnerability studies and how this method can be used in other regions. We have responded to each comment in

turn below in bold.

General suggestions

The first suggestion is regarding the absence of a clear drought definition. The sentences in the Introduction [R36-37] are insufficient in describing what kind of 'numerous droughts' China has experienced, and how this study is related to drought studies in China or globally. The specific naming of the 2000-01 event and the frequent occurrence of drought [R118] calls for a rigid definition of a drought. Later in the Introduction, it only becomes clear that this study focuses on meteorological and (soil moisture) agricultural droughts. In my opinion, this should have been stated earlier and clearer.

**Thank you for your suggestion. We will extend the drought definition in the Introduction to clarify these points in the revised manuscript. We will also add some explanation of the kind of drought China has experienced, and also add relevant literature to explain the relationship between this study and other related drought studies.**

In addition to this, the manuscript gives little explanation of previous meteorological or agricultural drought events, even though multiple authors have described droughts in China on both national and regional level (Wu, et al. 2001; Zou, et al. 2005; Leng, et al. 2015, Xiao-jun, et al. 2012; Wang, et al. 2016). It would be beneficial to the manuscript to explore the link with previous studies and build on other national-scale drought studies to claim further implications of this study. For example, the presented dataset seems unique and unpublished, although the term 'China water resources bulletins' in Xiao-jun, et al. (2012) suggests that there are multiple sources of drought impact data. I would suggest that acknowledging of these relevant studies, as it helps to rightly place this new study in context of previous research and thereby support the claim of further implications of this study [R97-98 and R364-366]. In lines R364-366, it is stated that the method could be applied to other areas, although it remains unexplained how to do so. Results in Figure 3 and 4 suggests that the linking between drought impact data and climate indices is fruitful despite the large climate variability.

The results show the strong relation between SPEI6 and Drought suffering area (DSA), SPEI6 and drought impacted area (DIA), and yield reduction and NDVI. These relations could be explored further in the Discussion section (R364-366), if a rigid drought definition is applied and the findings are related to relevant studies. That would increase the outreach of the developed method and would therefore benefit the manuscript significantly. In other words, I would strongly recommend to 1) provide a definition of the studied drought events, 2) relate them to past events –strengthen objective 1- and 3) link the findings to other drought studies in China to show the relevance of this study. Given the current structure of the introduction, I would expect that these suggestions would strengthen both the first, second and sixth paragraph [R87-91].

We thank the reviewer for these comments and we fully agree with him on these points. As the reviewers said, readers will have a lot of confusion without a clear definition of drought. Therefore we have added the definition of drought in the introduction. We explained what kind of drought China has experienced, and also added relevant literature to explain the relationship between this study and other related drought studies.

We have added the relationship analysis with the previous related studies in nation scale, such as the historic drought events, drought indices, and how this method can be applied in other regions. The details are as follows.

' In China, many indices were used for types of drought monitoring, such as Palmer drought index, SPEI, SPI, China-Z index, relative soil moisture, vegetation indices and remote sensing indices (Hong et al., 2001;Wang and Chen, 2014;Wu et al., 2012;Yanping et al., 2018), which found that serious drought events occurred in 1972, 1978, 1991 , 1999, 2000 and 2006. Based on previous drought studies, SPI, SPEI, soil moisture and NDVI were selected in this research. '

'The methods used here can be applied in other areas to better understand drought impacts and drought vulnerability, since similar data (e.g. drought impacts, meteorological data) can be collected in other regions.'

In addition to suggestion 1, I would suggest to include relevant drought studies in China that have explored a meteorological index (Wu, et al. 2001), agricultural droughts (as referenced) and water resource management strategies (Xiao-jun, et al. 2012). The current overview given in paragraph 6 does not reflect the full spectrum of relevant studies, hence I would strongly suggest for a thorough review of relevant studies in China to emphasise the link between previous studies and these findings. These studies have also performed analysis using multiple sources of information and could therefore strengthen the second paragraph in the discussion R321-335

We thank the reviewer for this important comment. We made some comparison with other related studies in data, method and results. According to your suggestion, we have added the comparison of relevant literature in the introduction and discussion section.

'In drought monitoring, the index selected in this study is similar to the method in Leng et al. (2015), where the SPI, standardized runoff index (SRI) and standardized soil moisture index (SSWI) were selected to assess droughts from meteorological, agricultural, and hydrologic perspectives. In terms of drought impact data, Xiao-jun et al. (2012) collected drought affected and damaged area, losses in food yield from China water resources bulletins, which is the secondary data.'

'The above results are also in general agreement with Hao et al. (2011), their study used a higher temporal and spatial resolution for drought impacts. It collected 10-day affected crop area data to assess drought risk in China at county unit. Their result shows that West Liaohe Plain has a high risk, northwestern part of Liaoning province are located at West Liaohe Plain.'

This is consistent with existing research by (Yan et al., 2012;Zhang et al., 2012), which established a drought risk assessment index system to assess drought risk in northwestern Liaoning. In Zhang et al. (2012), indices such as precipitation, water resources, crop area, irrigation capacity and drought resistance cost are used to measure drought risk, result shows that high drought risk was identified in Fuxin, Chaoyang and Shenyang.

The second suggestion concerns another definition; the use of the term vulnerability and the vulnerability assessment. In the Introduction, the relationship between drought indices, impact, and vulnerability is mentioned [R73-74], although in that same paragraph there is very little background given on the term 'drought vulnerability' or the chosen approach of this study. Later in the manuscript, R147-150, it becomes evident that vulnerability factors are related to agricultural productivity. It would strengthen the claim of 'developing a drought vulnerability evaluation' [R97], if the choice of vulnerability factors was justified earlier in the manuscript, perhaps supported using relevant literature to drought vulnerability.

We will add more background information on drought vulnerability as a term, and improve the definition in the introduction. As for the vulnerability factors, as most of the impacts available for Liaoning Province relate to agriculture and the rural economy, we selected the vulnerability factors to reflect this, also taking guidance from the studies of Junling et al., 2015 and Kang et al., 2014. We will add this information to the revised manuscript.

Also we added more explanation in how do we quantitatively assess drought vulnerability.

The vulnerability factors themselves (Table 2) require some additional adjustment in my opinion. Currently, these factors do not relate to normal conditions, or below-normal conditions, i.e. drought conditions. The standardisation in R215-220 shows that vulnerability factors are a ratio that is relative to the maximum amount measured for an unknown time scale. It remains unknown how these factors are measured or would change over time and since these vulnerability factors are not given as a reduction from normal conditions, it remains unclear to the reader how they represent vulnerability. Without the full understanding of the vulnerability factors, the impact of Figure

8 is limited, as these vulnerability levels do not indicate vulnerability as such, solely a reduction from the maximum number. For example, it remains unclear what 'most vulnerable to' implies in Figure 8, and more explanation is required to understand which factors are in or excluded for which cities. If so, it would require some more explanation regarding the rationale behind these 'most vulnerable to' factors. Once the vulnerability factors are converted into a deviation from the long-term mean (or however a drought is defined), the combined effect of these factors would become clearer. I do not expect the results to change, although the factors will and potentially show the deviation from the mean (or normal) conditions and therefore emphasise the change during droughts. The results

might show an amplified effects, which will help to strengthen the claim in R288-289. Along the same lines, I would also change the PHD, NLH and DELA into a percentage or ratio that relates to normal conditions. In the conclusion, relatively strong statements in R288-289 suggest that there is increasing drought vulnerability. However, from Figure 8 or Figure 7, it remains unclear how the vulnerability changes in Liaoning province, and these suggestions might aid the general analysis of the vulnerability factors.

Thank you for your comments. The manuscript may not be clear here before. The vulnerability factor is relative static to a specific city, which is the characteristics of the city. The maximum value refers to the maximum value among 14 cities in Liaoning Province, not the maximum value of a city for a period. In this paper, we ignore the changes of vulnerability for a period time, mainly emphasizing the difference of the vulnerability factors between cities.

We assumed that these factors was static for a period of time and that are collected by local government.

For each city, we analyze the relationship between vulnerability (measured by types of drought impacts at the same drought severity) and vulnerability factors to explore the contribution of vulnerability factors to each type of drought impacts.

For example when SPEI6 is equal to -1.5, the regression results show that yield loss due to drought is 5 thousand ton in Chaoyang whilst it is 1 thousand ton in Huludao. It means that in the term of the yield loss due to drought, Chaoyang is more vulnerable than Huludao.

Thank you for your suggestion using the percentage of the drought impacts. It would be better if drought impacts are display with percentage. However some drought impacts are difficult to convert to percentage, such as economic losses [0.1b], it's difficult for us to get a value to be divided to obtain the percentage. Similarly, due to the total number of livestock is not available in each city, we can't get the percentage too. Above all, it is difficult to show the impacts in the form of percentage.

The third suggestion is regarding the varying time scale of the multiple datasets. The presented data and analysis combine multiple datasets of varying quality and sources into one product. That in itself is a fine bit of work, although I would suggest to show the applied time scale in the correlation analysis and in the random forest modelling. It is not a major concern, but it would strengthen the manuscript to frame a defined study period that matches all data analysed in the correlation analysis, i.e. 1990-2013. In R191-192 and in R211-212, a short statement is written regarding the limitations of the soil moisture data and the NDVI data. Perhaps, an additional note regarding the applied study period is best written here.

Thank you very much for your suggestion. We agree with the reviewer that we need to add the period. In the method section, we have added the period of time series for each analysis.

For consistency, I would also emphasise the applied time period for the random forest algorithm (as introduced in the third section of the Methods). In the current manuscript, the applied time period remains unknown for the Random Forest algorithm. In fact, to enhance clarity, a brief summary of the work of Bachmair, et al. (2016) would be beneficial for readers that are less familiar with this algorithm. Again, minor adjustments in
the text would enhance the understanding of applied methods and therefore improve the manuscript.

We will add some more background to the approach in the revised manuscript and as mentioned above clarify the time periods over which the random forest analysis was conducted. We have added some explanation of MSE%, also we've added an example to explain the MSE% to make it clear.

Last correction I would suggest is the text along with Figure 5. In the figure, the coloured matrix gives the mean squared error in percentage. Firstly, I would strongly suggest to adjust the colour scheme to allow a non-experienced reader to see the difference between positive and negative percentage changes. Secondly, the change in MSE % suggests given a certain impact factor changes the error. If I read it correctly in R209, the change shows how much the accuracy decreases given the effect of the variable. This can be explained better than just one line of text, as a positive change in MSE% would imply not more MSE, but a more accurate model. Given the colour scale and the limited information available, the findings are somewhat hidden in this Figure despite the quality of the work. Hence, I would argue to change the colour scale accordingly and elaborate more in the text, i.e. give some examples.

We thank the reviewer for this comment and we fully agree with him on this point. According to reviewer's suggestion, we tried other color schemes, including blue, gray, brown, etc. to highlight the difference between positive and negative values. Finally, according to the visual effect and other references, we changed the color scheme.

Specific comments

Regarding the aggregation of impact data to an annual time scale, I would suggest to dedicate a short paragraph in the Discussion [R340-349] to show if results change for a multi-year drought (2000-01) or for a one year drought (2009). You might be better placed to identify example drought events, but it would strengthen statements in R334-346.

[Figure]

Thank you for your suggestion. We have added some explanations about the difference between the results of multi-year drought and single year drought.

The NDVI results show both positive and negative correlations. In lines R334-335, it is stated that this could be due to diversity of land cover, but given the detailed vulnerability factors, I would assume that there could be a more elaborate answer to these correlations. It would strengthen the discussion section to highlight some of correlations to plausible explanation regarding, e.g. land cover, change of cropping, use of perennial crops, etc.

We thank the reviewer for this important comment. According to reviewer's suggestion, we will add some detailed explanation. In other studies NDVI is mainly used to identify vegetation (agriculture) impacts. In this research, affected human and livestock are also collected to measure drought impacts.

Given the large spatial and temporal variability in precipitation [R108-110], it would be relevant to indicate the difference in water resources in addition to the variability in precipitation. The current annual average volume [R114-115] might not be relevant to drought conditions or vulnerability to droughts. The deviation from normal (annual average conditions) is relevant for drought research, how these droughts relate to the already water stressed areas might be detected by the climate indices.

We agree with the reviewer that the spatial and temporal variability in water resources need to be detected, we added the distribution characteristics of water resources in Liaoning Province.

The skewed distribution of water resources might play a part in the results of the DSA and DIA. It would be useful to indicate the deviation from mean, or the difference in source of water, rather than the amount that is available [R336-339]. In 358-360, the source and diversity of water sources is again linked to the vulnerability. This statement could benefit from an example case, where the source or variability in water resources indeed increased the vulnerability, as your results show.

We thank the reviewer for this important comment. We have added some examples the difference of water sources between NLH and PHD. To illustrate the importance of different water sources as you suggest.

Change the layout of Table 1 so that the vulnerability factors are easier readable. This would shift the focus from being on the spatial variability (which would be better shown in a map than a table) to the different vulnerability factors

We thank the reviewer for this comment and we fully agree with him on this point. It would be better shown in a map than a table. We have tried to plot a map to display the vulnerability factors. There will be ten maps (one for each type of drought impact), more space needed for these maps. Also the threshold of each type of drought impacts need be identified. Therefor we used table to show the vulnerability factors.

Depending on the applied drought definition (see general comment 1), mark this in Figure 2 to show the identified droughts. That will make it easier for the reader to deduct how the authors come to their findings in R128.

We agree with the reviewer and we added the definition of drought as suggest. In Figure 2, we use the SPEI as an example to illustrate the historical drought situation in Liaoning Province.

Change the current volumes and amount in [0.1b] yuan of drought impact in percentages. For a reader that is not familiar with current production levels in Liaoning province, it is hard to grasp the loss of 1.89 million tons, or the impact of an economic loss 1.87 billion yuan when the normal conditions are not provided [R120-121]

Thank you for your suggestion. We agree with the reviewer that it is more readable using percentages. Some drought impacts are difficult to express as a percentage, such as economic losses [0.1b], it's difficult for us to get a value to be divided to obtain the percentage. Also, due to the total number of livestock is not available in each city, we can't get the percentage value. As similar with Yield loss due to drought. Above all,

it is difficult to show the impacts in percentage since "normal conditions" means there is no drought occurred with no drought impacts.

Repeat the abbreviations in Table 1 in the text and perhaps in Figure 3,4, and 5. The abbreviations are used throughout the result sections, but are only fully explained in Table 1. I would suggest to repeat the abbreviations in the text to enhance the readability. For example, include (DI) in R124 and (SDI) R218. Same for the vulnerability factors NLH and PHD [R223]. It would be better to first write them full, before abbreviating even though these are given in table 1

We thank the reviewer for this comment and we fully agree with him on this point. For Figure 3, figure 4 and figure 5, we have added the full name of the drought impact rather than abbreviations to enhance the readability.

Also we have used the full terms in Discussion and Conclusion when it is first appear.

Need to support claims in drought mitigation strategies (e.g. sinking(?) more wells to enhance resilience to drought) R362-363.

We agree with the reviewer and we have added more drought mitigation strategies.

Could the authors clarify that the drought vulnerability map [R361] is indeed Figure 8?

Based on the results of the vulnerability analysis, figure 8 shows which cities have a higher vulnerability to which drought impacts. It displays that which city is vulnerable to what kinds of drought impacts.

Other than in the abstract (R29-31), no findings are related to future applications for other regions in China. Please revise the abstract, as these statements cannot be supported given the current manuscript.

We thank the reviewer for this comment and we revised the abstract as suggest.

In R124 the meteorological data is introduced, I assume that this data is obtained from all stations in Figure 1, please indicate which stations were use, or refer to the figure in

R124. The same holds for the soil moisture data in [R129]

Thank you for your suggestion. We added explanatory text to explain all the sites in Figure 1 were used.

Explain the difference between the applied SPEI using the log-logistic probability distribution (Yu, et al. 2014) [R165-166] and the often used method of Vicente-Serrano, et al. 2010).

Thank you very much for your suggestion, we changed the references.

Timeframe in R231 is 1990-2013 not 2016. Or, perhaps there is a mistake in the Figure 3 legend

Yes, thank you for your suggestion. We have corrected it.

Rephrase line 158-159

Yes we rephrased the sentence to make it clearer.

Rephrase line 286-288

Yes we have rephrased the sentence to make it clearer.

Rephrase line 314-316

Yes, the sentence has been corrected and made clearer.

Add 'of RF' in R356

Corrected.

Rephrase line 358-360

Rephrased.

References

Hao, L., Zhang, X., and Liu, S.: Risk assessment to China's agricultural drought disaster in county unit, Natural Hazards, 61, 785-801, 2011.

Yan, L., Zhang, J., Wang, C., Yan, D., Liu, X., and Tong, Z.: Vulnerability evaluation and regionalization of drought disaster risk of maize in Northwestern Liaoning Province, Chinese Journal of Eco-Agriculture, 20, 788-794, 2012.

Zhang, J. Q., Yan, D. H., Wang, C. Y., Liu, X. P., and Tong, Z. J.: A Study on Risk Assessment and Risk Regionalization of Agricultural Drought Disaster in Northwestern Regions of Liaoning Province, Journal of Disaster Prevention & Mitigation Engineering, 2012.

Hao, L., Zhang, X., and Liu, S.: Risk assessment to China's agricultural drought disaster in county unit, Natural Hazards, 61, 785-801, 2011.

Yan, L., Zhang, J., Wang, C., Yan, D., Liu, X., and Tong, Z.: Vulnerability evaluation and regionalization of drought disaster risk of maize in Northwestern Liaoning Province, Chinese Journal of Eco-Agriculture, 20, 788-794, 2012.

Zhang, J. Q., Yan, D. H., Wang, C. Y., Liu, X. P., and Tong, Z. J.: A Study on Risk Assessment and Risk Regionalization of Agricultural Drought Disaster in Northwestern Regions of Liaoning Province, Journal of Disaster Prevention & Mitigation Engineering, 2012.

Hao, L., Zhang, X., and Liu, S.: Risk assessment to China's agricultural drought disaster in county unit, Natural Hazards, 61, 785-801, 2011.

Bao, G., Liu, Y., Liu, N., and Linderholm, H. W.: Drought variability in eastern Mongolian Plateau and its linkages to the large-scale climate forcing, CLIMATE DYNAMICS,

Jinhua, C., Weiguo, Y., Ruina, L., Wei, Y., and Xi, C.: Daily standardized antecedent precipitation evapotranspiration index(SAPEI) and its adaptability in Anhui Province, Chinese Journal of Eco-Agriculture, 2019.

Yu, M., Li, Q., Lu, G., Wang, H., and Li, P.: Development and application of a short-

/long-term composited drought index in the upper Huaihe River basin, China, 369, 103-108, 2015.

---

## Referee Report (RR1)

The paper identifies the major drought events that affected Liaoning province (China) and determines which are the best indices to monitor drought in that area, which cities are more vulnerable to droughts and which vulnerability factors exacerbate drought consequences. Various drought indices (SPI, SPEI, NDVI and soil moisture) were computed and the relationship between drought indices and drought impacts was investigated using Pearson correlation coefficient and random forest models. Drought impacts were retrieved from the State Flood Control and Drought Relief Headquarters (SFDH) databases and were grouped in 8 categories. Drought vulnerability of the various cities of Liaoning province was derived based on the results of the correlation analysis and the random forest models. It was found that the most severe drought events in Liaoning province occurred in 2000-2001 and 2009. Among the considered indices, SPEI6 exhibited the strongest correlation with drought impacts. The cities located in the North Western part of Liaoning province are the most vulnerable to drought and, as it can be expected, the amount of crop cultivated area is a strong predictor of drought vulnerability.

The authors have addressed my previous comments, and the scientific quality of the paper has improved.

However, English language is still poor, the paper is not easy to read, and many sentences are not clear enough to the reader. Just to mention a few examples:

*Lines 27-29: The term drought is defined as meteorological, agricultural, hydrological, social and ecological drought. Meteorological drought is defined as a deficit of rainfall for a period in respect to the long term mean (Le Houérou, 1996). Then other types of drought can follow this definition.*

*Lines 46-47: In China, many indices were used for types of drought monitoring, such as Palmer Drought Severity Index (PDSI), SPEI, SPI, China Z index, relative soil moisture and remote sensing indices.*

*Lines 93-94: In Hao et al. (2011), drought impact s only measured by affected crop area in a 10 day time step in 93 county level.*

*Lines 301-303: Considering the various impacts, Chaoyang, Jinzhou, Tieling, Fuxin and Shenyang had the highest drought vulnerability, which are all located in the northwest part of Liaoning province.*

*Lines 373-374: The above results are also in general agreement with Hao et al. (2011), their study used 10 day affected crop area data as the drought impacts to assess drought risk in China in county unit.*

Therefore, an improvement in the English language is necessary to enable readers to fully understand the manuscript and appreciate the results reported.

---

## Referee Report (RR2)

[referee-annotated manuscript omitted]

---

## Author Response (AR2)

**Manuscript nhess-2019-310 "Linking drought indices to impacts to support drought risk assessment in Liaoning province, China" – Response to Reviewer 1**

The authors have addressed my previous comments, and the scientific quality of the paper has improved.

However, English language is still poor, the paper is not easy to read, and many sentences are not clear enough to the reader.

Just to mention a few examples:

**We thank the referee for the feedback to our manuscript, we will improve the English in the revised manuscript, paying particular attention to the examples given below.**

Lines 27-29: The term drought is defined as meteorological, agricultural, hydrological, social and ecological drought. Meteorological drought is defined as a deficit of rainfall for a period in respect to the long term mean (Le Houérou, 1996).

Then other types of drought can follow this definition.

**We will improve the English and clarity of this sentence in the revised paper.**

Lines   46-47: In China, many indices were used for types of drought monitoring, such as Palmer Drought Severity Index (PDSI), SPEI, SPI, China Z index, relative soil moisture and remote sensing indices.

**We will improve the English and clarity of this sentence in the revised paper.**

Lines 93-94: In Hao et al. (2011), drought impact s only measured by affected crop area in a 10 day time step in 93 county level.

**We will improve the English and clarity of this sentence in the revised paper.**

Lines 301-303: Considering the various impacts, Chaoyang, Jinzhou, Tieling, Fuxin and Shenyang had the highest drought vulnerability, which are all located in the northwest part of Liaoning province.

**We will improve the English and clarity of this sentence in the revised paper.**

Lines 373-374: The above results are also in general agreement with Hao et al. (2011), their study used 10 day affected crop area data as the drought impacts to assess drought risk in China in county unit.

**We will improve the English and clarity of this sentence in the revised paper.**

**Manuscript nhess-2019-310 "Linking drought indices to impacts to support drought risk assessment in Liaoning province, China" – Response to reviewer 2 (Veit Blauhut)**

The authors did a very good job in revising their manuscript. I added few suggestions and marked some words / lines to be rephrased.

**Thank you for your comments on the manuscript – we have responded to each of your comments below in bold text.**

Line 20 'slightly'

**Answer: We will revise this sentence.**

Line 29 'Then other types of drought can follow this definition. '

**Answer: We will revise this sentence.**

Line 34 'means'

**Answer: We will revise this sentence.**

Line 35 'this'

**Answer: We will revise this sentence.**

Line 36 'Some'

**Answer: We will revise this sentence.**

Line 39 you might ref to un-isdr 2009 or Hagenlocher et al.2019 here.

**Answer: we will added the suggested reference.**

Line 52 'Drought indices are focus..'

**Answer: we replaced 'Drought indices are' with 'Drought monitoring '**

Line 55 'some other similar analysis'

**Answer: we have replaced 'some other similar analysis' with 'random forest modeling '**

Line 57 Good reference for the wealth of impacts, not so good for vulnerbility.

**Answer: we have replaced the reference.**

Line 63 'necessarily'

**Answer: We will revise this sentence.**

Line 67 please add the information pooled---agriculture, water supply…?

**Answer: we have replaced 'drought impacts statistics in every village' with 'drought impacts on agriculture, industrial economy, and water supply in every village. '**

Line 94 Again please explan what city leve means in China I guess it is prefectural level?

**Answer: yes, the cities are prefectural, we will make this clearer in the revised paper.**

L98-99 Please rephrase, there are more including vulnerability... you might check on hagenlocher et al. 2019; of casue you can use this as an example, thank you

**Answer: We will rephrase this sentence and add the suggested reference to the revised paper.**

Line 178 'The 12, 15, 178 18 and 24 months SPI and SPEI in ending December were analyzed with the annual drought impacts during 1990 to 2013. '

**Answer: We will revise this sentence**

Figure 7 bad resolution, you might consider to put them all in one an colorscheme them.

**Answer: We will improve the resolution and clarity of the figure – we initially had all the cities on one plot, but this was hard to see – particualy due to the high number of colours required.**

Line 311 But this shows you the quality of linkage between losses and index, but not the vulnerabiluty itself! Please rephrase.

**Answer: we have rephrased the sentence ' Since drought impacts are symptoms of vulnerability, it can be used to estimate vulnerability (Blauhut et al., 2015a). For a specific severity of drought…'**

Line 321 'more or less population'

**Answer: we have added the 'more'.**

Line 323 sorry I kind of was expecting a final vulnerability map here.

**Answer:Thanks for your suggestion, it would be more readable to show the linkage between the vulnerability factors and drought impacts. Therefor we have added the stepwise regression Standardized Coefficients of each model in Table 3.**

Line 325-329 first and second are same

**Answer: we have rephrased the sentence.**

Line 331 of what?

**Answer: we have rephase the sentence with '.to assess drought vulnerability of eight drought impacts in Liaoning province.' to make it clearer.**

Line 332-L333 how, why, and how did you include them yet?

**Answer: We have revised this paragraph to provide more exaplanation for how the impact data were used given**

**their**

Line 337 what did others find out on that?

**Answer: The results may change if we used multi-year drought impacts, as longer index accumulation periods may have a stronger correlation with multi-year drought impacts than single year drought impacts. Other research has noted the importance of multi-year droughts in very long instrumental and archive records for**

**Liaoning (Tang et al. 2019), suggesting this could be an avenue for further research (e.g. Tang et al. 2019)**

Line 341 thoughts and findings of other studies are missing here.

**Answer:    In Nam et al. (2012) study, effective drought management can be achieved using drought monitoring**

**if the current conditions can be assessed and be updated on the latest drought situation. Then, the annual soil moisture does not reflect current drought conditions at a specific time.**

Line 346-348 But why? I suggets irrigation is a major driver? Please explicityl dicuss this here at your exampes.

**Answer: We will expand this discussion in the revised paper.**

Line 356 I'm not fully sure, but this could also be an issue of "freedom" in the model, nummber of trees, overfitting?, number of max. layers!

**Answer: We replaced this sentence to be more inclusive of the range of limitations that may have determined the RF performance.**

Line 402    some

**Anwer: We will revise this sentence**

Line 411    you might inlcude a "Hit list" of most important vulneravility factors-----some thing readers really like. Not forgetting the vulnerabilty map

**Answer: Thanks for your suggestion, The "Hit list" of most important vulnerability factors are displayed in Table 3. We have also clarified which the most 'important' vulnerability factors are in the conclusion.**

**List of all relevant changes made in the manuscript**

Line 10    'recurring' →'recurring'

Line 11 'characterizing'→ 'characterising'

Line 14    we insert the sentence :'for indices used in monitoring activities. '

Line 18-22    'To achieve this we use independent, but complementary, methods (correlation and random forest analysis) to identify which indices link best to drought impacts for prefectural-level cities in Liaoning province, using a comprehensive database of reported drought impacts whereby impacts are classified into a range of categories. The results show that Standardised Precipitation Evapotranspiration Index with a 6-month accumulation (SPEI6) had a strong correlation with all categories of drought impacts, while Standardised Precipitation Index with a 12-month accumulation (SPI12) had a weak correlation with drought impacts.' → 'To achieve this we use independent, but complementary, methods (correlation and random forest analysis) to identify which indices link best to drought impacts for prefectural-level cities in Liaoning province, using a comprehensive database of reported drought impacts whereby impacts are classified into a range of categories. The results show that Standardised Precipitation Evapotranspiration Index with a 6-month accumulation (SPEI6) had a strong correlation with all categories of drought impacts, while Standardised Precipitation Index with a 12-month accumulation (SPI12) had a weak correlation with drought impacts.'

Line 22    we deleted the 'sightly'

Line 23 'the study.' → 'The results of this study'

Line 24 we added 'and'

Line 25 we added 'The study also demonstrates the potential benefits of routine collection of drought impact information on a local scale.'

Line 28    ' hazards, and can cause numerous and severe societal impacts.' → ' hazards, and can cause numerous and severe societal impacts.'

Line 29    ' and delayed with respect to the event; ' →    'are often have a delayed onset in relation to the start of the drought event;'

Line 30    'The term drought is defined as ' → ' There are a number of 'types' of drought (Wilhite and Glantz, 1985), such as'

Line 31 'Meteorological drought is defined as a deficit of rainfall for a period in respect to the long term mean (Houérou, 1996). (Maracchi, 2000)Then other types of drought can follow this definition.'→ 'As these rainfall deficits propagate through the hydrological cycle, the other drought types occur as deficits occur in river flows, soil moisture and groundwater. Eventually impacts become manifest on the environment and society. '

Line 31 'From spring 2000 to autumn 2001, Liaoning province experienced a severe drought, which captured a large amount of attention from stakeholders and caused serious impacts on many sectors because of the consecutive years of drought.'→ 'Liaoning province experienced a severe drought from spring 2000 to autumn 2001 which captured a large amount of attention from stakeholders and caused serious impacts as a result of the consecutive years of drought '

Line 38 'providing'→' and provides the'

Line 39 'Some'→' Some of'

Line 39 'on'→' on the'

Line 41 'characterize '→ 'characterise '

Line 44 'A wealth of drought indices have been used in the literature'→' There are a wealth of drought indices in the literature'

Line 45 'although predominantly for drought monitoring and early warning (e.g. the review of Bachmair et al. 2016b) rather than risk assessment. '→' however they have predominantly used for drought monitoring and early warning (e.g. Bachmair et al. 2016b) rather than drought risk assessment applications.'

Line 47 'Standardized'→ 'Standardised '

Line 49-50 'SPI for three or six months are used for agricultural drought monitoring while SPI values for 12 or 24 months are normally applied to hydrological drought monitoring'→' SPI accumulations for 12 or 24 months are often applied to monitor hydrological droughts '

Line 51 'many indices were used for types of drought monitoring'→' many indices are used for drought monitoring'

Line Line 54 we delete 'the'.

Line 55 emphasizs → emphasis

Line 56   Drought indices are focus on meteorological and agricultural drought monitoring in China. → Drought monitoring efforts in China tend to focus on meteorological and agricultural drought monitoring;

Line 57 . Based on previous drought studies, SPI, SPEI, soil moisture   → based on this and previous drought studies, the SPI

Line 59    established by a correlation or some other similar analysis (e.g. Bachmair et al. 2016a), can thus be used → established by statistical methods (e.g. Bachmair et al. (2016a)), can be used

Line 55 drought impacts provide a proxy for vulnerability by demonstrating adverse consequences of a given drought severity (Blauhut et al., 2015a).    →    drought impacts are 'symptoms' of drought vulnerability and provide a proxy for vulnerability appraisal by demonstrating adverse consequences of a given drought severity (Blauhut et al., 2015a).

  Line 65 analyze → analyse

Line 68 we insert the ' as a result of '

Line 71 formalized →formalised

Line 55 who collect drought impacts statistics in every village. → who collect information on drought impacts on agriculture, industrial economy, and water supply in every village.

  Line 80     linking generally→ generally linking

Line 81    – recent studies tend to be in Europe, utilizing the EDII. → there are several recent studies in Europe, utilising impact reports from the EDII

Line 85 However, they →    However, all four studies

Line 86-87, meaning that all drought impacts had an equal weight without considering the duration, intensity or spatial extent of the impacts.    →    the number of impacts or a combination of both. This means that all drought impacts had an equal weight without considering the duration, intensity or spatial extent of the individual impacts.

Line 87-88, In contrast, Karavitis et al. (2014) analysed drought impacts transformed into monetary losses to measure drought impacts in Greece; however, it is    →    Karavitis et al. (2014) described drought impacts transformed into monetary losses to measure drought impacts in Greece. However, it

Line 92 we added 'and'

Line 93    analyzed → analysed

Line 96-101    Xiao-jun et al. (2012) collected annual drought affected area, damaged area, and annual losses in food yield in nation level from China Water Resources Bulletins to explore the water management strategies during droughts. In Hao et al. (2011), drought impacts were only measured by the affected crop area at the 10-day time step at the county level. In our research, eight types of drought impacts are collected to measure drought impacts in at the city unit (i.e. prefectural) level in Liaoning province, including the drought affected area, damaged area and yield loss, but also drought impacts on humans, livestock and the agricultural economy. →Xiao-jun et al. (2012) collected annual drought affected area and damaged area, annual losses in food yield in nation level from China water resources bulletins, which is the secondary data, to explore the water management strategies. In Hao et al. (2011), drought impacts is only measured by affected crop area in county level. In our research, eight types of drought impacts are collected to measure drought impacts in prefectural level in Liaoning province, which include not only drought affected area, damaged area and yield loss, but also drought impact on human, livestock and agricultural economy.

Line 102-104 In summary, previous studies have been focused on linking impacts to only one characteristic of drought (such as intensity, duration of occurrence) with most focusing on meteorological drought and agricultural impacts. But with the exception of Blauhut et al. (2015a) and Blauhut et al. (2016), there is little application of the results to drought vulnerability assessments. → In summary, previous studies have focused on linking impacts to only one characteristic of drought (such as intensity or duration of occurrence) with most focusing on meteorological drought and agricultural impacts with little application of the results to drought vulnerability assessments, with the exception of Blauhut et al. (2015a), Blauhut et al. (2016) and Hagenlocher et al. (2019), for example.

Line111 we delete ', based on the correlation analysis from objective 2,'

Line 117, 122 'cities'→ 'prefectural '

Line 124 we insert 'in Liaoning province '

Line 128. Spring maize → '; spring maize'

Line 137 we deleted the sentence ' representative '

Line 142-143 frequency domain reflection soil moisture sensors, which are based on the principle of electromagnetic pulse. Soil moisture data were not available between November and February at most stations due to freezing conditions. → frequency domain reflectometry soil moisture sensors Soil moisture data were not available at most stations between November and February due to freezing conditions.

Line 145 Monthly MODIS NDVI data → Monthly MODIS Normalised Difference Vegetation Index (NDVI) data

Line148 'centralized'→ 'centralised
'

**Line 155** Table 1: The eight drought impact categories for Liaoning province used in this study collected by the SFDH.→Table 1: The eight drought impacts used in this study collected by the SFDH for Liaoning province.

Line159 'Liaoning province '→ 'Liaoning Province

Line160 ', shown in Table 2'→ .'and are shown in Table 2 for each city unit.

Line161 'Vulnerability factors for each city in Liaoning province '→ 'Vulnerability factors for Liaoning province

Line165 and in the whole manuscript 'Standardized→ 'Standardised

Line167 'Organization'→ 'Organisation

Line171 'normalization'→ 'normalisation

Line 174 Here→ In this study

[revised manuscript text omitted]

Table 3 we insert the 'Standardised Coefficients'

We added the Scatterplots as figure 9.

[revised manuscript text omitted]